# Towards Unified Dynamic Face Landmark Detection

## Abstract

Although advancements in face landmark detection (FLD) methods continue to push performance boundaries, they overlook two major functional limitations: (1) different network parameters need to be trained independently for each "$N$-point" benchmark dataset, and (2) a model trained on an "$N$-point" dataset reliably outputs only the $N$ landmarks. In our work, we first conceptualize Face Part-Anchored Landmark Positions (FPALPs), wherein each landmark is treated as a progression value between zero (start) and one (end) along a face part's contour. Every landmark can be expressed in the FPALP format, irrespective of its source dataset, hence unlocking the ability to unify all "$N$-point" datasets into a single dataset. Secondly, we represent each landmark with an FPALP-based query, refine it progressively with a cross-modality decoder, and predict its coordinates based on the final representation. Our approach, called Unified Dynamic FLD, embodies these two design choices and streamlines the landmark detection pipeline by enabling (1) a single model to learn on any number of "$N$-point" datasets, and (2) yield any number of specific landmark predictions by loading the designated landmark queries at runtime. Extensive experiments carried out on several benchmark datasets demonstrate that our approach can achieve the above benefits while performing competitively with existing SOTA methods.

## 1 Introduction

Face landmark detection (FLD) aims to predict the coordinates of predefined landmarks in facial images. Facial landmarks provide rich and diverse visual cues regarding face shape, face-part positions, and pose information. These are essential for many downstream tasks such as 3D face reconstruction (Wood et al., 2022; Cai et al., 2021), face recognition (Juhong and Pintavirooj, 2017; Sarsenov and Latuta, 2017), face expression recognition (Munasinghe, 2018; Ngoc et al., 2020), and more recently facial beauty predictions (Bougourzi et al., 2022; J. Iyer et al., 2021) and face make-up try on (Marelli et al., 2022; Kips et al., 2021; Li et al., 2019; Sanapala and Angel Arul Jothi, 2024).

Despite being at the core of numerous applications, FLD algorithms suffer from major inherent drawbacks at both the training and prediction stages due to a *rigid adherence to the landmark layout defined by the training dataset*. Facial images are annotated with different landmark definitions across datasets such as AFLW (Zhu et al., 2015) (19/21 points), 300W (Sagonas et al., 2013) (68 points), and WFLW (Wu et al., 2018) (98 points). Generalizing, we denote an FLD dataset that defines a unique face layout of $N$ landmarks as the term "$N$-point" dataset. Prior works (Xia et al., 2022; Zhou et al., 2023; Huang et al., 2021; Li et al., 2022) have advanced FLD performance on these datasets by training on them *individually* using *separate* backbones and/or regression heads, and designing the networks to *output only the dataset-specific $N$ points*. We denote the above as the separate model and common backbone paradigms (see Figure 1) and investigate their demerits in detail below.

Theoretically, each "$N$-point" dataset can specify facial landmarks according to mutually exclusive semantic definitions. Here, the specialist nature of the separate model paradigm may outweigh the benefits of a model that was trained on multiple datasets through the common backbone paradigm, as only low-level features might be shared. In reality, we observe that *this assumption does not always hold true*. As an example, in Figure 2a, we overlay the landmark predictions output by SLPT (Xia et al., 2022), a state-of-the-art FLD method, that was trained separately on three benchmark datasets; AFLW19 (Zhu et al., 2015), 300W (Sagonas et al., 2013), and WFLW (Wu et al., 2018). We make

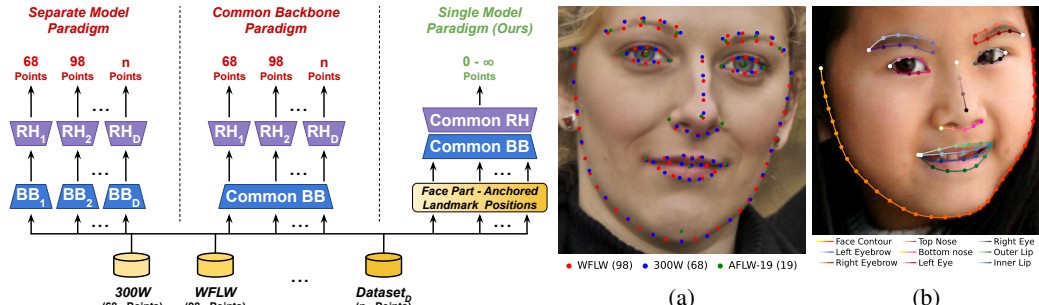

Figure 1: A comparison of the end-to-end training pipeline of prior works' separate model and common backbone paradigms to the single model paradigm implemented by our Unified Dynamic Face Landmark Detection method. $BB, RH$, and $D$ denote backbone, regression head, and number of datasets, respectively. Based on Face Part-Anchored Landmark Positions, our network can train on the combination of multiple "$N$-point" datasets and execute an unlimited number of landmark predictions.

Figure 2: (a) An overlay of the facial landmarks in the AFLW (19-point), 300W (68-point), and WFLW (98-point) formats. The landmark definitions across different datasets are observed to be non-mutually exclusive and strongly semantically related via face parts. (b) Landmarks (excluding pupils) of the WFLW format expressed as Face Part-Anchored Landmark Positions. Each gradient curve transitioning from white to a darker colour indicates the progression from the start to the end of a face part boundary.

two critical observations: (1) facial landmark annotations are semantically anchored to face parts such as eyes, lips, nose, etc., and (2) are often defined to be evenly spaced along a face part boundary (Wu et al., 2018; Yang et al., 2024). These cause the landmarks in the different "$N$-point" datasets to be *non-mutually exclusive and strongly semantically related*. Based on these observations, we conceptualize Face Part-Anchored Landmark Positions (FPALPs), in which each facial landmark is first associated with one or more distinct face parts and then assigned a value between 0 and 1 designating a progression point between the start and end of the face part boundary, respectively. We illustrate FPALPs in Figure 2b, wherein most facial landmarks of the WFLW (Wu et al., 2018) format are anchored to 9 distinct face parts. By indexing facial landmarks as FPALPs calculated on the union of all landmark definitions across the different benchmark datasets, we enable *unified FLD: the ability of an FLD model to be trained end-to-end on the combination of all the considered datasets.*

As noted earlier, during inference, FLD methods trained on an "$N$-point" dataset outputs only $N$ facial landmarks. Such output rigidity is non-optimal for downstream applications like face direction estimation (Al-Nuimi and Mohammed, 2021; Souley Dosso et al., 2022) and FLD stabilization in videos (Jin et al., 2020; Wu et al., 2021) that may utilize only a few sparse facial landmarks, and restrictive for applications like face image animation (Zhao et al., 2021) that require a higher density of accurate facial landmarks. Although higher facial landmark density can be naïvely achieved using interpolation methods, the output accuracy is dependent on a higher $N$ since face parts have non-linear shape. To this end, we construct facial landmark queries on demand using the combination of their FPALPs and the text embedding of the containing face parts, and feed them to a cross-modality decoder-regressor to enable *dynamic FLD: the ability of an FLD model to output the predictions of only the queried landmarks.*

Revisiting Figure 1, our Unified Dynamic FLD, which is founded on the concept of FPALPs, executes a *single model paradigm* that can be trained on the combination of diverse "$N$-point" datasets, and can yield any number of specific facial landmark predictions at inference time. In Table 1, we compare the efficiency of the single model paradigm of our method with the separate model and common backbone paradigms executed by prior work, when trained on $D$ number of unique "$N$-point" datasets. Visibly, our method is the *most efficient* since it is agnostic to $D$ on all the considered factors, and is the *most versatile* since it offers demand-specific landmark throughput.

Table 1: An efficiency comparison of different face landmark detection paradigms. $D$ denotes the number of unique "$N$-point" datasets. $\mathcal{B}, \mathcal{H}$, and $E_Q$ denotes the backbone, regression heads, and landmark query encoder respectively.

| FLD Paradigm | Efficiency | | | |
| | Training Cycles | Inference Calculation | Storage Parameter | Landmark Throughput |
| --- | --- | --- | --- | --- |
| Separate Model | $D$ | $D\mathcal{B} + D\mathcal{H}$ | $D\mathcal{B} + D\mathcal{H}$ | $N$ |
| Common Backbone | $D$ | $1\mathcal{B} + D\mathcal{H}$ | $1\mathcal{B} + D\mathcal{H}$ | $N$ |
| Single Model (Ours) | $1$ | $1\mathcal{B} + 1\mathcal{H} + 1E_Q$ ($E_Q << \mathcal{H}$) | $1\mathcal{B} + 1\mathcal{H} + 1E_Q$ ($E_Q << \mathcal{H}$) | $0 - \infty$ |

Our contributions and their benefits are summarized below:

1. We propose the Face Part-Anchored Landmark Positions (FPALPs), an intuitive representation of face landmarks that are evenly distributed on well-defined face part curves. The FPALP format is universal and allows for compatibility with all existing and future datasets.

2. To the best of our knowledge, our work using FPALPs is the first to enable, *without auxiliary dataset information*, Unified FLD: the ability of a model to be trained end-to-end on the fusion of multiple "$N$-point" datasets. We demonstrate increased model generalization by training on a larger, more diverse combined dataset offering higher landmark heterogeneity through the unification of various "$N$-point" formats.

3. We propose a novel FPALP-based landmark queried regressor to enable Dynamic FLD, i.e., unlimited on-demand landmark prediction without network retraining.

4. We demonstrate through extensive experiments that our work not only unlocks the numerous benefits of Unified Dynamic FLD but also achieves competitive performance compared to existing state-of-the-art methods on several benchmark datasets.

## 2  RELATED WORK

**Targeting Fundamental Performance Improvements.** Recent face landmark detection (FLD) methods can be categorized into direct coordinate regression methods (Li et al., 2022; Xia et al., 2022; Li et al., 2020) and heatmap-based regression methods (Huang et al., 2021; Zhou et al., 2023; Kumar et al., 2020). While each approach has advantages and disadvantages, they target different challenges to achieve performance improvements. AnchorFace (Xu et al., 2020) proposes a split-aggregate strategy using anchor templates to tackle landmark uncertainty in large pose faces. ADNet (Huang et al., 2021) and STARLoss (Zhou et al., 2023) address the semantic ambiguity in landmark annotations by suppressing the associated disentangled loss component for landmarks with an anisotropic distribution. DTLD (Li et al., 2022) and SLPT (Xia et al., 2022) adaptively leverage the underlying inter-landmark structural relationship to improve localization performance, especially on occluded landmarks. Meanwhile, PIPNet (Jin et al., 2021) performs simultaneous heatmap regression and offset predictions to speed up inference while achieving competitive localization accuracy. *Orthogonal to these efforts*, our work aims to deliver the aforementioned unified and dynamic FLD properties to induce *robustness and versatility at the system level*.

**Approaches to Ameliorate the FLD Pipeline.** Prior works have also surfaced the issues of immiscibility of the various "$N$-point" annotation schemes across datasets (Wu et al., 2018; Yang et al., 2024), and the infeasibility to infer landmarks beyond those $N$ defined by the training dataset (Yang et al., 2024; Chandran et al., 2023). LAB (Wu et al., 2018) represented facial structure using 13 boundary lines, theorized that facial landmarks across datasets can be interpolated within these lines, and performed landmark regression using the common backbone paradigm. LDDMM-Face (Yang et al., 2024) assigned landmarks on mean face templates to semantic boundary curves called flows and used flow-wise deformation layers to predict the final landmarks. The limited adaptation between different annotation schemes was achieved through affine transformation between the source and target mean faces. FreeEnricher (Huang et al., 2023) employs a patch refinement network operates on contextual patches centered on interpolated face landmarks along a given face part curve and predicts offsets to align them with the true face part boundary. Since the enrichment process is decoupled from the base landmark prediction network, its efficacy is highly dependent on the accuracy of the base landmark predictions that define the initial face part curve. Recently, CLD (Chandran et al., 2023) proposed a pipeline that ingests a facial image and arbitrary 3D query locations on a canonical face shape to output the corresponding and possibly continuous 2D landmark coordinates. Although CLD could be trained with multiple datasets, its success is highly dependent on a large collection of densely annotated face datasets having 3D canonical landmark mappings. While a more detailed comparison is provided in Appendix A.5, we briefly contrast it here to highlight our single-model paradigm, which trains end-to-end with only sparsely annotated 2D landmark datasets and performs dynamic and direct inference to any arbitrary landmark format *without manual transformations*. Furthermore, our Face Part-Anchored Landmark Position-based landmark queries are *easily interpretable* and allow for unconstrained interaction with text-based or agentic downstream applications.

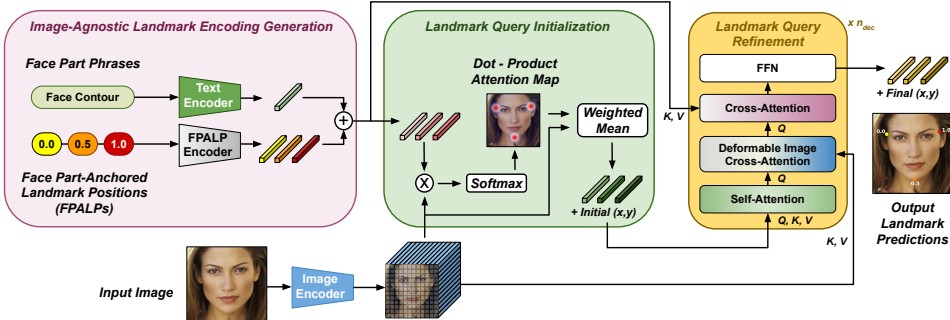

Figure 3: An overview of our proposed framework. First, we associate user-defined face parts to the required landmarks and calculate their Face Part-Anchored Landmark Positions (FPALPs). The FPALPs and the face parts' text are encoded and aggregated to yield the image-agnostic landmark encodings. The facial image's visual features are then conditioned on these encodings to output the initial landmark queries and coordinate predictions. Lastly, a cross-modality decoder block iteratively refines the landmark queries and coordinate predictions to output the final values.

**Generalist Face Models.** Another line of research aims to simultaneously perform facial tasks such as landmark detection, age/gender/head-pose estimation, and face parsing using multi-task learning. Early works like HyperFace (Ranjan et al., 2016) and AIO (Ranjan et al., 2017) utilized multi-scale features from various CNN layers and executed upto 7 face tasks at once using task-wise heads. Recently, FaceXFormer (Narayan et al., 2024) and Faceptor (Qin et al., 2024) treated face tasks as tokens in transformer-based architectures (Vaswani et al., 2017) containing unified task and pixel decoders. These on-demand task-expandable generalist face models train on the fusion of diverse task datasets. However, for the FLD task, they still train separately on the "$N$-point" datasets and yield only a fixed $N$ output. Our unified dynamic FLD method can be *readily integrated* into existing generalist face models to streamline their FLD division.

## 3 METHODOLOGY

Our proposed Unified Dynamic Face Landmark Detection framework is inspired by Grounding DINO (Liu et al., 2023) and is illustrated in Figure 3. First, we introduce Face Part-Anchored Landmark Positions (FPALPs), a supplementary representation of facial landmarks from the viewpoint of face part boundaries. Next, we describe how we construct image-agnostic landmark encodings using FPALPs and combine them with facial image features sourced from an image encoder to initialize the landmark queries and the primitive coordinate predictions. Lastly, we elucidate the process of iterative query refinement to yield the final landmark representations and coordinate predictions.

**Face Part-Anchored Landmark Positions (FPALPs).** As prior work (Wu et al., 2018; Yang et al., 2024) have noted and illustrated by us earlier in Figure 2a, facial landmarks specified by benchmark datasets that we consider, i.e., AFLW (Zhu et al., 2015), WFLW (Wu et al., 2018), and 300W (Sagonas et al., 2013), are bound to face part boundaries in an evenly spaced manner. To leverage this observation, we conceptualize Face Part-Anchored Landmark Positions (FPALPs). Here, each landmark is associated with one or more containing face parts and is represented as a progression value between 0 and 1 denoting its fractional position within the containing face part curve. Since each "$N$-point" dataset can define its face template with different landmark layouts and different start and end positions for the various face parts, we first create a unified face template by taking the union of the face templates of all datasets. Formally, we denote the face template for dataset $D_i$, out of $D$ considered datasets, having $N_i$ number of landmarks, as $T_{D_i}$, and the unified face template as $T_U = T_{D_1} \cup T_{D_2} \cup ... \cup T_{D_D}$. $T_U$ consists of $N_U$ number of landmarks clusters each of which indicates a landmark's proximity across the $D$ datasets. While this may seem inexact, we observed a clean alignment between the face templates resulting in tight proximal landmark clusters having an average intra-cluster distance of 2.22 pixels averaged over all face parts.

We split $T_U$ into $P$ face part templates, $T_U = T_{P_1} \cup T_{P_2} \cup ... \cup T_{P_P}$, each consisting of member landmarks to represent user-defined face part curves such as left/right eye(brow), face contour, inner/outer lip, etc. Face part curves can be open (e.g., nose bridge, face contour) or closed (e.g., eyes,

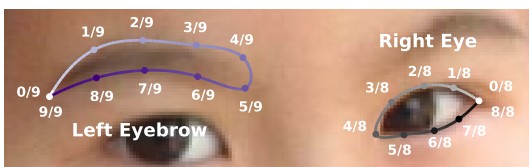

Figure 4: Illustration of the construction of Face Part-Anchored Landmark Positions (FPALPs) for the landmarks of the left eyebrow and the right eye closed curve face parts.

lips). For closed curve face parts, we create a copy of the starting landmark in the curve sequence and signify it as the ending landmark for that sequence. In Figure 4, we exemplify the following FPALP formulation using the left eyebrow and right eye face part curves. For a landmark $l$ positioned at $pos_{l,p}$ within a sequence of $N_p$ landmarks that compose the face part $p$ with template $T_p$, we define the FPALP of $l$ as $FPALP_{l,p} = pos_{l,p}/(N_p - 1)$.

**Image-Agnostic Landmark Encodings.** To achieve dynamic face landmark detection, we represent target landmarks as landmark queries. To this end, we construct initial image-agnostic representations which conceptually capture the landmarks to be queried. Firstly, we encode FPALPs using a simple MLP with ReLU activation. Next, we input the face part name into a lightweight pretrained text encoder to get its textual representation. Finally, we derive the image-agnostic landmark encodings as the summation of the encoded FPALPs and the face part textual representations. Formally, for a landmark $l$ in the face part $p$, the image-agnostic landmark encodings $E_{IA}^{l,p}$ are derived as:

$$E_{FPALP}^{l,p} = \mathrm{MLP}(FPALP_{l,p}), \quad E_{text}^p = \mathrm{Enc}_{text}(p), \quad E_{IA}^{l,p} = E_{FPALP}^{l,p} + E_{text}^p \quad (1)$$

where $E_{FPALP}^{l,p}, E_{text}^p, E_{IA}^{l,p} \in \mathbb{R}^d$, and $d$ is the encoding dimension. In lieu of $\mathrm{Enc}_{text}$, we could use learnable embeddings to yield the face part representations. We hypothesize that pretrained text encoders are more superior since they may already encode the semantics of facial layouts. In Sec. 4.2, we compare both the options and corroborate that using pretrained text encoders is the better choice.

**Landmark Query Initialization.** Effective initial landmark queries should capture the required landmarks' proximity specified by their semantic definitions. To this end, we condition the facial image's visual features with the image-agnostic landmark encodings in the following manner. First, we utilize a pretrained image encoder to output the facial image features $E_I \in \mathbb{R}^{H_{E_I} \times W_{E_I} \times d}$, where $(H_{E_I}, W_{E_I})$ represents the spatial resolution of the image features. Let $G_{E_I} = \{(x_{i,j}^c, y_{i,j}^c)\}_{i=0,j=0}^{H_{E_I}, W_{E_I}} \in \mathbb{R}^{H_{E_I} \times W_{E_I} \times 2}$ represent the grid of the image-space center coordinates corresponding to $E_I$. Next, we derive the attention map $A$ of the visual features with respect to the required image-agnostic landmark encodings $E_{IA} \in R^{L \times d}$, where $L$ denotes the number of landmarks to be queried, as $A = \mathrm{Softmax}(E_I \cdot E_{IA}^T)$, where $A \in \mathbb{R}^{H_{E_I} \times W_{E_I} \times L}$ and the softmax is applied along the $H_{E_I} \times W_{E_I}$ dimension. Here, $A$ reflects the activation of the visual regions that correspond to the required landmarks' image-agnostic landmark encodings. We obtain our initial landmark queries $Q_0 \in \mathbb{R}^{L \times d}$ and initial coordinate predictions $C_0 \in \mathbb{R}^{L \times 2}$ by taking the weighted mean of $E_I$ and $G_{E_I}$ using the attention map $A$, respectively. Formally, given grid center coordinates $(x_{i,j}^c, y_{i,j}^c) \in G_{E_I}$, for a required landmark $l \in [0, L)$ with a corresponding attention map $A^l \in R^{H_{E_I} \times W_{E_I}}$, the initial landmark query $LQ_0^l$ and initial coordinate prediction $C_0^l$ are derived as:

$$LQ_0^l = \sum_{i=0,j=0}^{H_{E_I}, W_{E_I}} A_{i,j}^l \cdot E_{I_{i,j}}, \qquad C_0^l = (\sum_{i=0,j=0}^{H_{E_I}, W_{E_I}} A_{i,j}^l \cdot x_{i,j}^c, \sum_{i=0,j=0}^{H_{E_I}, W_{E_I}} A_{i,j}^l \cdot y_{i,j}^c) \quad (2)$$

**Landmark Query Refinement.** We employ a cross-modality transformer decoder, as depicted in the third block of Figure 3, to iteratively hone the landmark queries and the predicted coordinates. This block consists of $n_{dec}$ decoder layers, the first of which consumes $LQ_0$ and $C_0$, while the later layers consume the output of the previous layers to implement iterative refinement. At layer $dec_i$, we first execute self-attention on the landmark queries $LQ_{dec_i-1}$ to exploit the inter-landmark dependencies. Then, we deploy a deformable attention (Zhu et al., 2021) layer that consumes the locations $C_{dec_i-1}$

and performs targeted cross-modality attention between the image features and the queries from the previous step. To reinforce the alignment between the queries and the semantic definitions of the required landmarks, we execute a cross-attention layer between the queries from the previous step and the image-agnostic landmark encodings $E_{IA}$. Finally, we deploy a feed-forward network to yield the decoder layer's query output $LQ_{dec_i}$, operate an MLP on it to derive the coordinate offsets with respect to $C_{dec_i-1}$, and calculate the coordinate predictions as $C_{dec_i}$. For brevity, we assume the reader to be aware of the transformer-related notations and formulations, and elucidate the above process using simplified equations as below:

$$LQ_{dec_i}^{SA} = \text{SelfAttn}(LQ_{dec_i-1}, LQ_{dec_i-1}, LQ_{dec_i-1}) \tag{3}$$

$$LQ_{dec_i}^{DICA} = \text{DeformableAttn}(LQ_{dec_i}^{SA}, E_I, E_I, C_{dec_i-1}) \tag{4}$$

$$LQ_{dec_i}^{CA} = \text{CrossAttn}(LQ_{dec_i}^{DICA}, E_{IA}, E_{IA}) \tag{5}$$

$$LQ_{dec_i} = \text{FFN}(LQ_{dec_i}^{CA}) \tag{6}$$

$$C_{dec_i} = C_{dec_i-1} + \text{MLP}(LQ_{dec_i}) \tag{7}$$

where $dec_i \in [1, n_{dec}]$ and the first three inputs to the layers in Equation 3-5 respectively assume the roles of query, key, and value in the attention mechanism.

Given the ground truth coordinates of the required $L$ landmarks $C_{GT} \in \mathbb{R}^{L \times 2}$, we supervise both our intermediate and final coordinate predictions $C_{dec_i}$ where $dec_i \in [0, n_{dec}]$ using the Wing Loss (Feng et al., 2018) as $\mathcal{L} = \sum_{dec_i=0}^{n_{dec}} \text{WingLoss}(C_{dec_i}, C_{GT})$.

## 4 EXPERIMENTS

**Datasets.** We train and evaluate our framework on three benchmark datasets: AFLW (Zhu et al., 2015), 300W (Sagonas et al., 2013), and WFLW (Wu et al., 2018). AFLW focuses on coarse annotations for in-the-wild images and comprises of 20000 training and 4386 test facial images, each annotated with 19 landmarks. 300W is collected from five facial datasets and contains 3148 training and 689 test facial images, each annotated with 68 landmarks. The test set is further divided into common (554 images) and challenging (135 images) subsets. WFLW is collected from WIDER Face (Yang et al., 2016) with an emphasis on challenging poses, expressions, and occlusions. It consists of 7500 training and 2500 test images, each annotated with 98 landmarks. For cross-dataset evaluation, we consider COFW (Burgos-Artizzu et al., 2013), which contains 507 test images each annotated with 29 landmarks, COFW68 and WFLW68, the 68 landmark variants whose face template matches that of 300W. Collectively, these datasets provide images with diverse levels of expression, pose, and occlusion, making them effective to evaluate a model's generalization ability.

**Implementation Details.** Facial images from all datasets are cropped using the given bounding boxes and resized to either $224 \times 224$ (ViT-B) or $256 \times 256$ (ResNet) depending on the image encoder. Following prior works (Jin et al., 2021; Li et al., 2022; Qin et al., 2024), bounding boxes are enlarged by 10% to include more contextual information. Data augmentation methods including random rotations ($\pm 15°$), scaling ($\pm 20\%$), horizontal flipping, and translation ($\pm 10$ pixels), are employed to improve model robustness by simulating real-world variability.

We employ the lightweight pretrained SentenceBERT (Reimers and Gurevych, 2019) as the face part text encoder, FaRL (Zheng et al., 2022) pretrained ViT-B (Dosovitskiy et al., 2021) or ResNet (He et al., 2016) as the facial image encoder, 3 decoder layers ($n_{dec}$), each with 8 attention heads, and a model-wide feature dimension $d = 256$. During image cross-attention, 4 features per head are sampled from each level of the image feature maps for each query. We train the model end-to-end on an NVIDIA A100 GPU (40GB) for 32 epochs, with a batch size of 16, using the Adam optimizer with a learning rate of $10^{-4}$ and a weight decay of $10^{-5}$. The learning rate is lowered to $10^{-5}$ from the 25th epoch. The image and text encoders are trained at a tenth of the running learning rate. Details on the dataset sampling strategy can be found in Appendix A.9.

**Evaluation Metrics.** Following prior works (Jin et al., 2021; Li et al., 2022; Xia et al., 2022), we evaluate the face landmark detection methods using the Normalized Mean Error (NME) percentage. NME measures the L2 distance between the predicted and true landmarks and is normalized by either the inter-ocular distance ($NME_{inter-ocular}$), which is used for evaluation on 300W and WFLW, or the diagonal distance of the facial bounding box ($NME_{diag}$), which is used for evaluation on AFLW.

Table 2: Comparison of our Unified Dynamic Face Landmark Detection approach with SOTA methods on the WFLW, 300W, and AFLW-19 datasets. Our method *enables fused dataset training and dynamic landmark prediction* with a negligible $0.05 - 0.1\%$ performance drop compared to SOTA methods on the full version of the datasets.

| Method | Method Type | Trained w/ Additional Datasets | Fused Dataset Training | Dynamic Landmark Prediction | WFLW Full | 300W Common | 300W Challenge | 300W Full | AFLW-19 Full |
|---|---|---|---|---|---|---|---|---|---|
| | | | | | NME$_{\text{inter-ocular}}$ ↓ | | | | NME$_{\text{diag}}$ ↓ |
| FaceXFormer (Narayan et al., 2024) | Generalist | ✓ | ✗ | ✗ | - | 2.66 | 4.67 | 3.05 | - |
| Faceptor (Qin et al., 2024) | | ✓ | ✗ | ✗ | 4.03 | **2.52** | **4.25** | **2.86** | **0.95** |
| PIPNet (Jin et al., 2021) | | ✗ | ✗ | ✗ | 4.31 | 2.78 | 4.89 | 3.19 | 1.42 |
| ADNet (Huang et al., 2021) | | ✗ | ✗ | ✗ | 4.14 | 2.53 | 4.58 | 2.93 | - |
| SLPT (Xia et al., 2022) | | ✗ | ✗ | ✗ | 4.14 | 2.75 | 4.90 | 3.17 | - |
| DTLD+ (Li et al., 2022) | Specialist | ✗ | ✗ | ✗ | 4.05 | 2.60 | 4.48 | 2.96 | 1.37 |
| STAR Loss (Zhou et al., 2023) | | ✗ | ✗ | ✗ | **4.02** | 2.52 | 4.32 | 2.87 | |
| Ours (ViT-B) | Specialist | ✗ | ✓ | ✓ | 4.07 | 2.59 | 4.50 | 2.96 | 1.04 |

## 4.1 Comparison with SOTA Methods

Our intention in providing quantitative comparisons with SOTA methods is to *demonstrate the competitiveness of our framework, not to establish new benchmarks.* Our main contribution is *the enablement of the unified and dynamic FLD features.* Additionally, as most prior works are not open-sourced and use different backbones, direct adaptation for fair comparison is challenging. Hence, for transparency, we cite their reported performance and explicate our model configuration.

**Individual Dataset Evaluation.** We compare our Unified Dynamic Face Landmark Detection (FLD) framework against SOTA methods and present the results in Table 2. We include the generalist approaches that use additional datasets from other tasks for reference purpose only. As seen from the table, our best model enables the training on the fusion of multiple training datasets and allows for dynamic landmark prediction, while performing *on-par* with prior works with a minor drop of $0.05 - 0.1\%$ in NME on the full version of all datasets. The negligible performance drop across multiple datasets further corroborates a *well-defined alignment* between the face templates of the considered datasets and proves that Face Part-Anchored Landmark Positions are *conceptually applicable* to the FLD task.

**Cross-Dataset Evaluation.** To verify the generalization ability of our approach, we conduct a cross-dataset evaluation on the COFW68 and WFLW68 datasets using our model trained only on the 300W dataset, and present the results in Table 3. Our method with the ResNet backbones fare approximately on par with SOTA on the 300W and COFW68 datasets. Using the ViT-B backbone, we demonstrate robustness by significantly improving performance on the challenging WFLW68 dataset, which includes facial images with extreme poses, expressions, occlusions, and makeup.

Table 3: Cross-dataset evaluation comparison. Models are supervised only on 300W.

| Method | 300W | COFW68 | WFLW68 |
|---|---|---|---|
| | NME$_{\text{inter-ocular}}$ ↓ | | |
| LAB(Wu et al., 2018) | 3.49 | 4.62 | - |
| AVSw/SAN(Qian et al., 2019) | 3.86 | 4.43 | - |
| DAG(Li et al., 2020) | 3.04 | 4.22 | - |
| PIPNet(Jin et al., 2021) | 3.36 | 4.55 | 8.09 |
| DTLD (Li et al., 2022) | 3.07 | **4.42** | 7.23 |
| Ours (ResNet18) | 3.17 | 4.88 | 7.36 |
| Ours (ResNet101) | 3.13 | 4.8 | 7.22 |
| Ours (ViT-B) | **3.03** | 4.46 | **6.11** |

Table 4: Ablation study on training datasets as a cross-dataset evaluation. * indicates exclusion of undefined landmarks not defined in the template.

| Training Datasets | AFLW-19 | 300W | WFLW | COFW | WFLW68 | COFW68 |
|---|---|---|---|---|---|---|
| | NME$_{\text{diag}}$ ↓ | NME$_{\text{inter-ocular}}$ ↓ | | | | |
| 300W | 2.25* | 3.04 | 6.47* | 3.85* | 6.11 | 4.46 |
| WFLW | 2.44* | 4.14 | 4.12 | 3.74 | **3.93** | 4.73 |
| 300W + WFLW | 2.47* | 3.03 | 4.16 | 3.74 | 4.46 | 4.39 |
| 300W + WFLW + AFLW-19 | **1.04** | 2.96 | 4.07 | 3.57 | 4.44 | **4.30** |

## 4.2 Ablation Studies

Our model is trained on a fusion of AFLW19, 300W, and WFLW, and we conduct a dataset ablation study to assess the contribution of each dataset to overall performance (Table 4). The study evaluates the impact of training on salient dataset combinations and tests on individual datasets. Notably, when the evaluation dataset uses a different "$N$-point" template than those seen during training, the setting is effectively *near zero-shot*, as many Face Part-Anchored Landmark Positions (FPALPs) in the target dataset are unseen. Our approach is the first to conduct such cross-template evaluations without resorting to manual interpolation techniques. The results indicate that training on all datasets combined yields the best performance across most datasets, except for WFLW68, where the best performance is achieved by training solely on WFLW. We attribute this to the reduction of non-critical landmarks in the transition from the 98-point to the 68-point template and the dilution of

challenging samples when additional datasets are introduced. It is essential to acknowledge that the effect of incorporating new training datasets can vary based on alignment between the distribution and label quality of the training and evaluation datasets. The observed performance gains from training on datasets with diverse face templates suggest that exposure to different FPALPs enhances the model's ability to effectively represent face part curves and generalize across varied facial structures and ambient conditions.

**Choices for Face Part Representation.** In this study, we investigate the impact of choosing how face parts are represented to yield $E_{text}^p$ in Equation 1. We present two options: (1) training learnable embeddings, or (2) leveraging the output of pretrained text encoders. The former might seem as the default option given the limited amount of face parts that can be encoded. We contend that, although simpler, training with *learnable embeddings may not capture the nontrivial semantics of facial structure*, such as the relative positions of face parts, the inter-face part relationships during facial expressions (e.g., the squinting of the eyes and broadening of the lips during a laugh), and interactions with makeup and accessories. We postulate that text encoders that are trained on diverse corpora encode these intricacies. In Figure 5, we compare the training curve plots of the model when using learnable embeddings versus SentenceBERT (Reimers and Gurevych, 2019), a lightweight pretrained text encoder, and in Table 5, we compare their performance at conver-

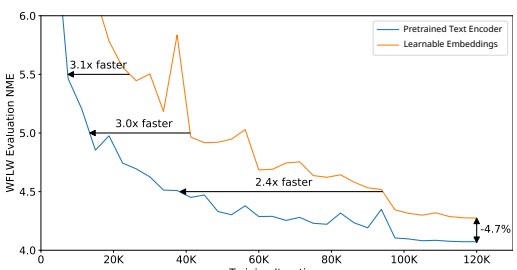

Figure 5: A comparison of training curve plots when using learnable embeddings vs. Sentence-BERT (Reimers and Gurevych, 2019) to represent face parts.

Table 5: Performance comparison using learnable vs. SentenceBERT (Reimers and Gurevych, 2019).

| Query Type | AFLW-19 $NME_{diag} \downarrow$ | 300W | WFLW $NME_{inter-ocular} \downarrow$ | COFW | COFW68 |
|---|---|---|---|---|---|
| Learnable | 1.09 | 3.04 | 4.27 | 3.79 | 4.36 |
| Language | **1.04** | **2.96** | **4.07** | **3.57** | **4.30** |

gence. The usage of SentenceBERT to represent face parts results in a faster convergence and a more performant model, thereby corroborating our earlier thesis and proving to be the superior choice over learnable embeddings.

**Impact of Image Encoder.** We analyze the impact of different image encoders, including ResNet18, ResNet101, and ViT-B, on the overall performance of our model as well as its generalization ability. In Table 6a, we display the results of our model with different backbones when trained on the fusion of the considered datasets. Although ViT-B proves to be superior on most of the evaluation datasets, it is noteworthy that both ResNets perform competitively, at a fraction of the size of ViT-B. This suggests that the availability of diverse "$N$-point" training datasets is of higher importance than the capacity of the image encoder to achieve an overall high-performing model. In Table 6b, we present the results of a cross-dataset evaluation conducted by training our model only on 300W, with different image encoders, and report its performance on 300W, COFW68, WFLW, WFLW68, and $WFLW_E$, whose face template contains only the 28 points that are absent/undefined in 300W. We observe that as the image encoder's capacity increases, the performance improves drastically on most datasets, especially on the $WFLW_E$ variant where our model executes *zero-shot* evaluation since the landmarks (and their corresponding FPALPs) are *unseen* during training. As we constrained our model training to only 300W, the result of this experiment suggests that the generalization ability of our model is dependent on the capacity of the image encoder.

Table 6: (a) Performance comparison of our model trained on the fusion of all considered datasets when using various image encoders. (b) Cross-dataset evaluation with our model trained only on 300W when using different image encoders. $WFLW_E$ refers to the WFLW dataset containing *only the 28 points* that are absent/undefined in 300W.

| Image Encoder | AFLW-19 $NME_{diag} \downarrow$ | 300W | WFLW $NME_{inter-ocular} \downarrow$ | COFW | COFW68 |
|---|---|---|---|---|---|
| ResNet18 | 1.07 | 3.00 | 4.39 | 3.77 | 4.53 |
| ResNet101 | 1.04 | **2.92** | 4.25 | 3.71 | 4.47 |
| ViT-B | **1.04** | 2.96 | **4.07** | **3.57** | **4.30** |

(a)

| Method | 300W | COFW68 | WFLW68 $NME_{inter-ocular} \downarrow$ | WFLW | $WFLW_E$ |
|---|---|---|---|---|---|
| ResNet18 | 3.17 | 4.88 | 7.22 | 7.60 | 7.96 |
| ResNet101 | 3.13 | 4.80 | 7.36 | 7.69 | 7.70 |
| ViT-B | **3.03** | **4.46** | **6.11** | **6.47** | **6.66** |

(b)

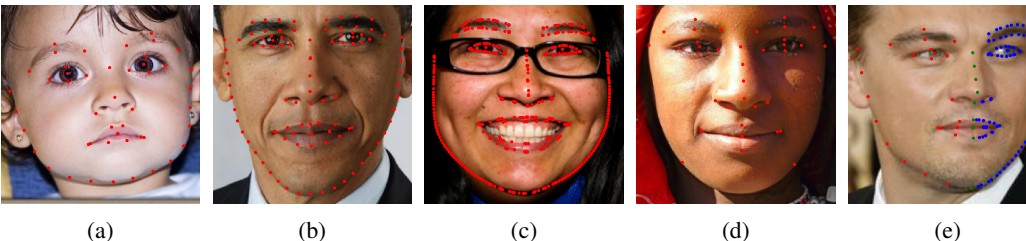

|  (a)  |  (b)  |  (c)  |  (d)  |  (e)  |

Figure 6: An illustration of the dynamic landmark prediction capability of our system. Images are selected from the WFLW (98-point) test set. We anchor the landmarks to the following face parts: left and right eyes, eyebrows, and pupils, inner and outer lips, face contour, nose bridge and boundary. For (e), we split the face contour, lips, and nose boundary into left, center, and right sub-parts. Landmark predictions per face part are depicted using (a)-(c) a granularity multiplier of 0.5, 1, and 4 respectively, (d) 4 landmarks per face part, and (e) a granularity multiplier of 0.5, 1, and 2 for left, center, and right sub-parts whose landmarks are color coded as red, green, and blue respectively.

### 4.3 DYNAMIC LANDMARK PREDICTION

In this section, we qualitatively assess the output of our Unified Dynamic Face Landmark Detection system. Using our model trained on the fusion of the considered datasets, we depict a variety of dynamic landmark prediction configurations on images from the WFLW test set in Figure 6. In contrast to prior works, which can only output a fixed landmark layout as in Figure 6b, our model can predict landmarks pertaining to user-selected face parts and, furthermore, at various granularities within (Figure 6a-6d) and across (Figure 6e) face parts, hence demonstrating its versatility and applicability to an assortment of downstream applications. We observe that landmark predictions for face parts with higher FPALP diversity, such as the face contour and the eyes, are more accurate than those with lower FPALP diversity, such as the nose boundary. A larger diversity of face templates within the combined training dataset increases the exposure to different FPALPs and empowers the model to accurately predict landmarks at higher granularities. Additionally, incorporating loss components that enforce appropriate distribution of landmark predictions across the input FPALPs could further enhance prediction quality – a direction we leave for future work.

## 5 DISCUSSION AND CONCLUSION

**Limitations.** We acknowledge the likelihood of an imprecise alignment of the individual datasets' face templates during the construction of the unified face template, which may hinder the scalability of the FPALP formulation. We note from the resiliency of our model, which is trained on the alignment of three different (14, 68, and 98-point) face templates, that only an approximate alignment is necessary for effective face part curve learning. In cases of large misalignment, new face parts can be defined to contain the introduced landmarks. Meaningful inter-face part relationships with the misaligned face parts can still be forged via the landmark query refinement process in our model.

**Future Work.** An extension of our work can be the construction of 2D FPALPs (detailed in Appendix A.8) to capture face part surfaces and leverage the query features to track facial artifacts like acne, moles, and wrinkle lines. Research on integration with vision-language and generalist face models may allow for text- or visual prompt-based face part creation, automated landmark-to-FPALP registration, and generation of the unified face template, leading to a more versatile face landmark detection component within a robust face analysis system.

**Conclusion.** In this paper, we present our Unified Dynamic Face Landmark Detection method, wherein landmarks are treated as progression points on user-defined face parts, allowing for end-to-end model training on the fusion of diverse "$N$-point" datasets and execution of unlimited on-demand landmark predictions. With a performance competitive with to SOTA methods, our simple, yet adaptable framework is positioned to meet the requirements of various downstream applications that depend on a wide range of precise face landmarks.

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

## A    TECHNICAL APPENDICES AND SUPPLEMENTARY MATERIAL

### A.1    RELEVANCE AND STRENGTH OF CONTRIBUTION

Our work addresses critical limitations in current 2D face landmark detection (FLD) methods and provides an efficient, semantically flexible alternative to both dense and traditional sparse approaches:

- **Unified Training Across Datasets.** We introduce the FPALP representation, which enables a single model to be trained across heterogeneous landmark templates without requiring 3D priors or costly alignment procedures. This overcomes the fragmentation seen in prior work, where separate models are typically needed for different datasets.

- **Dynamic, Semantic Landmark Prediction.** Unlike fixed-protocol models or dense outputs, our method supports flexible, part-based landmark queries. This design offers interpretability and adaptability for downstream tasks that demand only specific landmarks or face regions.

- **Purely 2D Supervision.** Our method operates entirely within the 2D domain, without relying on 3D annotations or model-based priors. This makes it more scalable and accessible in real-world applications where 3D data is limited or unavailable.

- **Improved Generalization and Regularization.** Training on diverse datasets with different landmark configurations serves as a natural regularizer, promoting robustness and reducing overfitting. The FPALP structure aligns these heterogeneous protocols into a unified representation that supports generalization to unseen templates.

- **Compatibility with Sparse-to-Dense Learning.** Our model supports zero-shot or near-zero-shot generalization across protocols. It can be trained on sparse landmarks and still perform well on denser configurations, laying the groundwork for bridging sparse and dense paradigms in a single framework.

- **Suitability for Low-Resource Deployment.** Dynamic 2D FLD is particularly advantageous for edge and mobile devices, where lightweight, semantically interpretable, and adaptable models are essential. Our method meets these requirements, while 3D-based approaches—due to their reliance on dense meshes, heavy computation, and 3D priors—are ill-suited for such environments. By avoiding these constraints, our framework provides a practical and efficient solution for real-world deployment.

### A.2    LIMITATIONS CONTINUED

From our ablation studies, we infer that the generalization ability of our model, which we define as its capacity to accurately predict landmarks at unseen Face Part-Anchored Landmark Positions (FPALPs) – is influenced by both the diversity of training dataset face templates and the range of facial and ambient conditions. Training on a broader variety of datasets with distinct landmark layouts, rather than simply increasing the number of datasets with similar layouts, is likely to yield a more generalizable model. However, as discussed in the dataset ablation section, incorporating additional training datasets may enhance overall generalization but could also diminish performance on specific evaluation datasets if the training and evaluation datasets differ significantly in facial and ambient condition distributions or label quality.

We also note that FPALPs are constructed using native dataset annotations. These annotations along face part boundaries often represent semantic progression points in 2D space. However, landmarks generated through interpolation techniques may not align with those predicted via evenly spaced FPALPs. For instance, consider a front-facing view of a person whose jawline narrows near the chin. Landmarks sampled along the true jaw contour may increase in density as they approach the chin. While these landmarks may appear equidistant in the profile view, they may not be evenly spaced in the frontal view. Consequently, we are unable to quantitatively evaluate our method on benchmarks that employ higher-density landmarks derived through interpolation, as it becomes challenging to objectively identify the cases we described above.

Lastly, we acknowledge that the choice of the training dataset and the text encoder could introduce biases or limitations when face part phrases are described using low-resource languages. In order to mitigate such biases and limitations, the definition of face part phrases should be standardized for a consistent interpretation across the different languages, and the text encoder would either need to

be trained or fine-tuned on the target languages. While this is a crucial consideration for real-world deployment in a global context, our current work limits its scope to the English language context, deferring inter-linguistic adaptations and broader cross-cultural considerations to future research.

## A.3 EXCLUSION OF DATASETS

Our framework is trained and evaluated on AFLW19 (Zhu et al., 2015), WFLW (Wu et al., 2018), and 300W (Sagonas et al., 2013), with additional evaluation conducted on COFW (Burgos-Artizzu et al., 2013) and its variants. However, we exclude COFW from the training set due to observed inconsistencies in annotation quality. Preliminary experiments indicated that including COFW not only degraded the overall performance across all datasets but also adversely impacted the quality of denser landmark predictions. We also do not evaluate on the Enriched 300W test set proposed in (Huang et al., 2023) as its annotations are derived through interpolation-based methods, as discussed in the A.2 above.

## A.4 CHOICE OF TEXT ENCODER

As discussed in the main paper, we employ SentenceBERT (Reimers and Gurevych, 2019) as the text encoder to generate face part representations. In Sec. 4.2, we detailed the rationale for selecting a language model output rather than a learnable embedding. Another plausible option was to use the FaRL (Zheng et al., 2022) text encoder, given that we already utilize its image encoder. Although FaRL was trained on LAION-FACE (Zheng et al., 2022), a dataset comprising facial image-text pairs, the textual descriptions predominantly consist of general attributes such as "smiling girl with party wig" or "the beautiful bride with the sunlight shining on her," rather than the specific face part intricacies discussed in Sec. 4.2. In contrast, SentenceBERT, having been pretrained on diverse and extensive textual corpora, demonstrated a superior ability to effectively encode these more detailed and nuanced characteristics of individual face parts.

While our study confirmed that using the FaRL text encoder yielded superior performance compared to generic learnable embeddings, our experiments ultimately revealed that SentenceBERT outperformed FaRL's text encoder for our specific task. This indicated that, for generating image-agnostic landmark encodings using face part phrases, a superior representation of the specific semantics of face parts is achieved by using a strong pretrained text encoder. Therefore, despite the potential benefits of image-text alignment, the pretrained, lightweight SentenceBERT proved to be the more effective choice for encoding face part phrases in our framework.

## A.5 DETAILED COMPARISON WITH CONTINUOUS LANDMARK DETECTION (CHANDRAN ET AL., 2023)

As outlined in Sec. 2, Continuous Landmark Detection (CLD) (Chandran et al., 2023) is a recent framework that takes as input a facial image and arbitrary 3D query locations on a canonical 3D face surface to output corresponding 2D landmark predictions. While CLD can be trained using existing 2D face landmark datasets, it requires a layout mapping to the 3D canonical surface, imposing a dependency on such mappings. In contrast, our Unified Dynamic FLD framework eliminates this dependency by deriving FPALPs directly from the native coordinate system of the dataset, enabling training on all native 2D FLD datasets without additional mappings.

Furthermore, CLD leverages 3D coordinates on the canonical face mesh as input queries, facilitating continuous landmark detection, and is a very valuable contribution, especially in applications where accurate dense coordinates are required to retrieve and characterize a facial surface. In contrast, our framework is designed to provide a more interpretable and semantically driven interface for querying FLD systems. Specifically, we construct queries based on text-defined face parts and semantic progression points along face contours. We envisage future FLD systems being queried using descriptive instructions, such as "Predict 10 coordinates from the left chin boundary to the end of the jawline," and position our framework to address such needs.

From an architectural perspective, CLD's query encoder processes 3D query locations on a canonical face mesh, whereas our framework encodes face part text and FPALPs. Additionally, while CLD's landmark predictor employs transformer layers to fuse the image encoder output and the 3D query encoding, our framework instantiates the initial landmark queries and coordinate predictions by

encoding the face part text and corresponding FPALPs and subsequently conditioning the image features. We then refine the queries and coordinate predictions using self-attention and cross-attention layers to produce the final landmark coordinates.

## A.6 DETAILED FPALP FORMULATION

In this section, we revisit and elaborate on the formulation of the Face Part-Anchored Landmark Positions (FPALPs). Referring to the FPALP formulation in the main paper, for a landmark $l$ positioned at $pos_{l,p}$ in a sequence of $N_p$ landmarks that composes the face part $p$ with template $T_p$, we denote the FPALP of $l$ as $FPALP_{l,p} = pos_{l,p}/(N_p - 1)$. As observed, the landmark layout pertaining to a face part defined by a dataset usually comprises of landmarks that are evenly distributed on the face part boundary. The unification of individual landmark templates of the various datasets into $T_U$ may render the collection of landmarks to be unevenly distributed along the face part boundary. To determine $pos_{l,p}$ of a landmark $l$ which originally belonged to the dataset $D_i$ with landmark layout $T_{D_i}$, we first derive the position of the face part's starting landmark in $T_{D_i}$ relative to the starting landmark of the face part in $T_U$, and then add to it the index of $l$ relative to the other landmarks of the face part in $T_{D_i}$. We express the above formulation for $pos_{l,p}$ as:

$$pos_{l,p} = \text{RelativePosition}(l_{start,p}^{T_{D_i}}, l_{start,p}^{T_U}) + index_{l,p}^{T_{D_i}} \tag{8}$$

The FPALP formulation normalizes progression along each face part from 0 to 1 in a dataset-agnostic manner. Crucially, the start and end points of a face-part curve in FPALPs are not fixed by any dataset; they are defined once by the practitioner when specifying the face-part phrase and its member landmarks. FPALPs then assign each (face part, landmark) pair a normalized position in [0,1] along that user-defined ordering. This means that the same physical landmark can legitimately receive different FPALP values under different, possibly overlapping, face-part phrases (e.g., "nose" = "nose bridge + left + right boundary of nose" vs "right nasal region" = "right boundary of nose + right nasolabial fold"), and differences in how individual datasets choose their "first" or "last" landmark on a contour do not constrain the unified representation.

## A.7 HANDLING UNDEFINED OR OCCLUDED FACE PARTS

Our current framework assumes that the queried face parts are explicitly defined in the training data. We acknowledge that parts that are heavily occluded or undefined poses a challenge and the impact would be dependent on the extent of visible visual context. To address such cases, the framework could be extended in future work to dynamically infer or adapt face part boundaries:

1. We can utilize the text encoder to parse face part descriptions into latent embeddings that can be aligned with image features. Soft spatial attention maps based on the introduced face parts can be used to approximate the boundaries of unseen face parts, even under occlusion. Such an extension would enable the model to infer FPALP-like progression values for novel regions by projecting the learned attention map onto surrounding anchor contours. Additionally, a dynamic part discovery module could be trained using contrastive losses to bind new textual descriptions to consistent visual patterns across samples. This could potentially enable open-vocabulary part generalization in FLD, which could be an exciting avenue for future work.

2. We can also leverage visibility annotations per landmark, such as those provided in the MERL-RAV (Kumar et al., 2020) dataset, to supervise the model in learning to selectively ignore occluded regions during training. This allows the framework to learn robust part representations even when portions of the face are not visible. Additionally, these visibility flags can be used to guide a gating mechanism or soft-attention masking module that modulates the contribution of occluded regions in the query or image features during inference, improving landmark prediction reliability under occlusion.

## A.8 DETAILED 2D FPALP PROPOSAL FOR FUTURE WORK

As discussed in the future work section, the proposed FPALPs can be extended to 2D space to facilitate further advancements. Currently, FPALPs are defined in 1D space, representing semantic

progression points along a face part curve. By treating face part curves as boundaries, 2D FPALPs can be defined along these boundaries, capturing semantic progression both horizontally and vertically, with either the x or y component as zero. Extending this further, regions within the face part boundary can be described using 2D FPALPs where both x and y components are non-zero. With only the face part boundary as input, weak supervision could be employed to predict 2D FPALPs for arbitrary points within the face part region. Thus, transitioning from 1D to 2D FPALPs shifts the representation from linearly traversing face part curves to encompassing face part surfaces.

While 1D FPALPs correspond to progression along a face part boundary, 2D FPALPs require a surface parameterization that maps internal face part regions to a normalized coordinate space. Constructing such mappings without dense annotations firstly requires us to define the boundary coordinates of each defined face part in both spatial dimensions and further necessitates the use of weak supervision to learn the face part surface. For example, given only the boundary of a region (e.g., the cheek or forehead), one could generate pseudo-ground-truth 2D FPALP labels using mesh-based interpolation to learn consistent internal representations across identities.

Incorporating 2D FPALPs would allow the model to reason over continuous face surfaces rather than just boundary curves, enabling richer spatial representations. This would benefit tasks such as facial expression analysis, where subtle shape changes within a region (e.g., the bulging of cheeks or wrinkling of the forehead) may not be captured effectively through sparse boundary points. By modeling internal face part regions with 2D FPALPs, the framework could localize and track deformations more precisely, potentially improving performance on downstream tasks requiring dense spatial awareness.

## A.9    Detailed Training Procedure

**Dataset Sampling.** As our model is trained on a fusion of multiple datasets, we apply dataset-level oversampling to ensure a balanced training distribution. Each training epoch includes approximately the same number of samples from each dataset, ensuring equal exposure to each $N$-point facial landmark template.

**Batch Sampling.** Since each dataset uses its own $N$-point template, all samples within a dataset share the same number of queried landmarks. For each training iteration, we randomly select (without replacement) one dataset and sample a batch (equal to the batch size) from it. This ensures consistent tensor shapes for landmark queries and avoids the need for jagged arrays.

## A.10    Explanation of Slight Performance Drop on WFLW68

We address this issue in L375–383 of the manuscript and expand on it here. As noted in L299–300, the WFLW dataset presents a wide range of challenges, including extreme poses, expressions, and occlusions. In our unified training setup, we apply dataset-level oversampling to maintain a balanced exposure across all datasets. However, because other datasets often contain less challenging samples, the model's exposure to difficult WFLW-specific cases is reduced. This can explain the slight performance drop on WFLW68. Importantly, while we observe a decrease in NME on the 68-point version of WFLW, we also observe a performance gain on the full 98-point format. This suggests that the model benefits from the additional diverse data, especially in handling the extra 30 facial points. In other words, the gain in the 30 additional landmarks outweighs the loss in the common 68, indicating that our method generalizes well overall when exposed to a wider variety of N-point formats.

## A.11    Face Template Alignment Statistics

As detailed in Sec. 3 of the main paper, the first step in the formulation of Face Part-Anchored Landmark Positions (FPALPs) is the synthesis of the unified face template $T_U$ through an alignment of the individual face templates of the considered datasets. We specified that the alignment of the face templates of AFLW19, 300W, and WFLW, resulted in tight proximal clusters having an average intra-cluster distance of 2.22 pixels averaged over all face parts. In Table 7 we expand this statistic by showing the per-face part mean intra-cluster distances of landmark clusters having at least two landmark members.

We theorize that these intra-cluster distances quantifies a blend of (1) semantic positioning inconsistency across multiple poses, arising from differences in how datasets define contour trajectories under foreshortening, profile rotations, occlusions and different facial expressions, and (2) subjective annotation noise at the annotator level. FPALPs operate under this noisy supervision yet still learn a stable "average" contour representation. A promising direction for future work is to explicitly model and decompose this noise, e.g., by estimating semantic curve variability separately from annotator-level deviations and training the model to minimize the former while remaining robust to the latter.

Table 7: Mean intra-cluster distance (in pixels) for the landmark clusters per face part during the alignment of the face templates of the AFLW19, 300W, and WFLW datasets, into a unified face template. A clean alignment is observed with the minimum, maximum, and mean values of the mean intra-cluster distance taken across the face parts as 1.51, 3.82, and 2.22 pixels respectively.

| Face Part | Mean Intra-Cluster Distance |
|---|---|
| face contour | 3.82 |
| left eyebrow | 2.11 |
| right eyebrow | 2.24 |
| nose bridge | 2.30 |
| nose boundary | 1.51 |
| left eye | 1.59 |
| right eye | 1.52 |
| outer lip | 1.82 |
| inner lip | 3.27 |
| left pupil | 2.12 |
| right pupil | 2.07 |

A.12 LANDMARK TO FPALP MAPPING

After we attain the unified face template $T_U$, we assign each landmark to one or more user-defined face parts and calculate its Face Part-Anchored Landmark Positions. We tabulate the result of this assignment for the AFLW, COFW, 300W, and 300W datasets in Tab. Table 8-11 respectively.

A.13 SHOWCASING DYNAMIC LANDMARK PREDICTION: ADDITIONAL VISUALIZATIONS

As in Fig. 6 within Sec. 4.3 of our main paper, we qualitatively assess the output of our Generalized Dynamic Face Landmark Detection system by depicting a variety of dynamic landmark prediction configurations on images from the WFLW test set in Figure 7. Further, in order to depict the robustness of our system, we visualize our landmark prediction configurations on the challenging cases of occlusion in Figure 8 and extreme poses in Figure 9 from the WFLW test set. From the visualizations, we observe that our framework is able to successfully reason and predict the most likely positions for the landmarks despite the (partial and complete) occlusion of face parts, atypical facial expressions, and extreme poses.

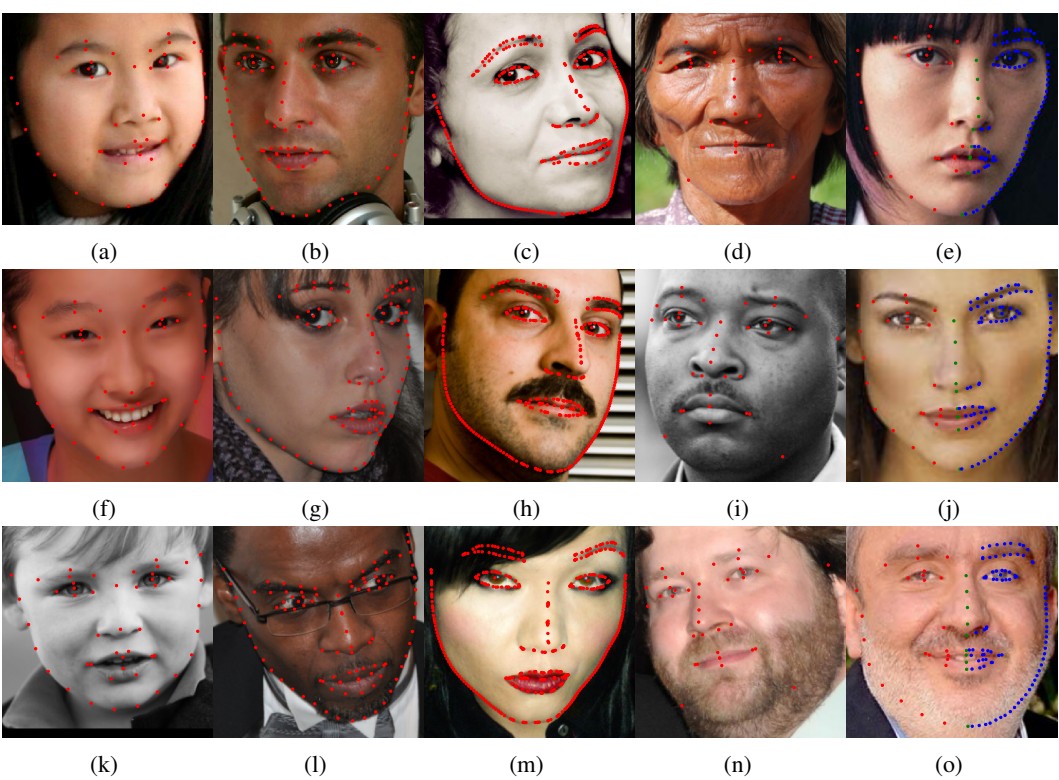

Figure 7: An illustration of the dynamic landmark prediction capability of our system. All images are selected from the WFLW test set which implements the 98-point face template. We anchor the landmarks to the following face parts: left and right eyes, eyebrows, and pupils, inner and outer lips, face contour, nose bridge and boundary. For (e), we split the face contour, lips, and nose boundary into left, center, and right sub-parts. Landmark predictions per face part are depicted using (a)(f)(k) a granularity multiplier of 0.5, (b)(g)(l) a granularity multiplier of 1, (c)(h)(m) a granularity multiplier of 4, (d)(i)(n) 4 landmarks per face part, and (e)(j)(o) a granularity multiplier of 0.5, 1, and 2 for left, center, and right sub-parts whose landmarks are color coded as red, green, and blue respectively.

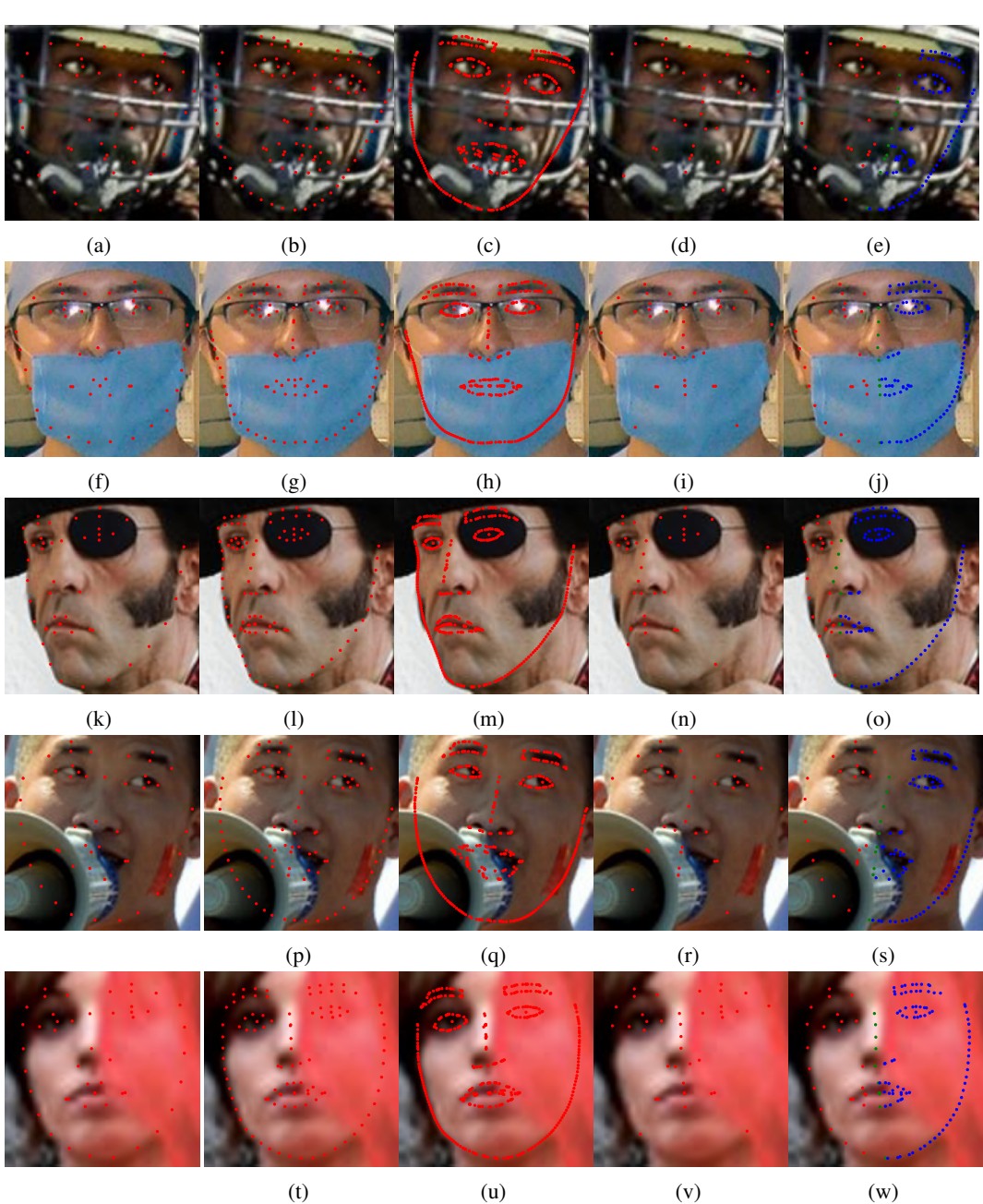

Figure 8: An illustration of the dynamic landmark prediction capability of our system on challenging occlusion cases from the WFLW test set. We anchor the landmarks to the following face parts: left and right eyes, eyebrows, and pupils, inner and outer lips, face contour, nose bridge and boundary. For (e), we split the face contour, lips, and nose boundary into left, center, and right sub-parts. Landmark predictions per face part are depicted using (a)(f)(k) a granularity multiplier of 0.5, (b)(g)(l) a granularity multiplier of 1, (c)(h)(m) a granularity multiplier of 4, (d)(i)(n) 4 landmarks per face part, and (e)(j)(o) a granularity multiplier of 0.5, 1, and 2 for left, center, and right sub-parts whose landmarks are color coded as red, green, and blue respectively.

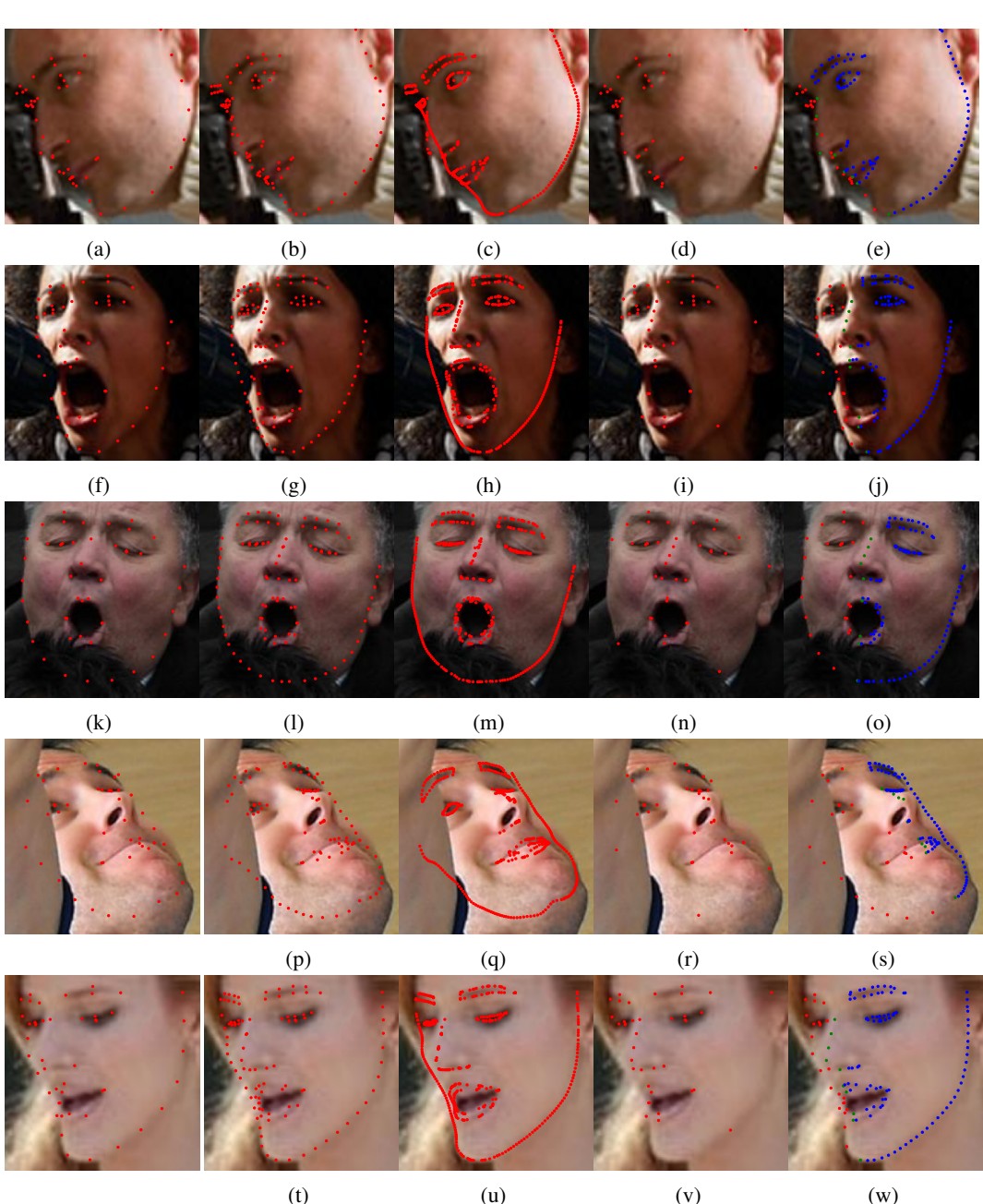

Figure 9: An illustration of the dynamic landmark prediction capability of our system on challenging extreme pose cases from the WFLW test set. We anchor the landmarks to the following face parts: left and right eyes, eyebrows, and pupils, inner and outer lips, face contour, nose bridge and boundary. For (e), we split the face contour, lips, and nose boundary into left, center, and right sub-parts. Landmark predictions per face part are depicted using (a)(f)(k) a granularity multiplier of 0.5, (b)(g)(l) a granularity multiplier of 1, (c)(h)(m) a granularity multiplier of 4, (d)(i)(n) 4 landmarks per face part, and (e)(j)(o) a granularity multiplier of 0.5, 1, and 2 for left, center, and right sub-parts whose landmarks are color coded as red, green, and blue respectively.

Table 8: Mapping of AFLW's 19 landmarks to their face parts and Face Part-Anchored Landmark Positions (FPALPs).

| Landmark ID | Face Part | FPALP |
|---|---|---|
| 1 | Face Contour | 16/32 |
| 2 | Left Eyebrow | 0/9 or 0/2 |
| 3 | Left Eyebrow | 4.5/9 or 1/2 |
| 4 | Right Eyebrow | 0/9 or 0/2 |
| 5 | Right Eyebrow | 4.5/9 or 1/2 |
| 6 | Middle of Left Eyebrow | 0/1 |
| 7 | Middle of Right Eyebrow | 0/1 |
| 8 | Nose Bridge | 3/3 |
| 9 | Nose Boundary | 0/6 |
| 10 | Nose Boundary | 6/6 |
| 11 | Left Eye | 0/6 |
| 12 | Left Eye | 3/6 or 1/2 |
| 13 | Right Eye | 0/6 |
| 14 | Right Eye | 3/6 or 1/2 |
| 15 | Outer Lip | 0/12 |
| 16 | Outer Lip | 6/12 |
| 17 | Middle of Mouth | 0/1 |
| 18 | Left Eye Pupil | 0/1 |
| 19 | Right Eye Pupil | 0/1 |

Table 9: Mapping of COFW's 29 landmarks to their face parts and Face Part-Anchored Landmark Positions (FPALPs).

| Landmark ID | Face Part | FPALP |
|---|---|---|
| 1 | Face Contour | 16/32 |
| 2 | Left Eyebrow | 0/9 |
| 3 | Left Eyebrow | 2/9 |
| 4 | Left Eyebrow | 1/2 |
| 5 | Left Eyebrow | 7/9 |
| 6 | Right Eyebrow | 0/9 |
| 7 | Right Eyebrow | 2/9 |
| 8 | Right Eyebrow | 1/2 |
| 9 | Right Eyebrow | 7/9 |
| 10 | Nose Bridge | 3/3 |
| 11 | Nose Boundary | 0/6 |
| 12 | Nose Boundary | 3/6 |
| 13 | Nose Boundary | 6/6 |
| 14 | Left Eye | 0/8 |
| 15 | Left Eye | 2/8 |
| 16 | Left Eye | 4/8 |
| 17 | Left Eye | 6/8 |
| 18 | Right Eye | 0/8 |
| 19 | Right Eye | 2/8 |
| 20 | Right Eye | 4/8 |
| 21 | Right Eye | 6/8 |
| 22 | Outer Lip | 0/12 |
| 23 | Outer Lip | 3/12 |
| 24 | Outer Lip | 6/12 |
| 25 | Outer Lip | 9/12 |
| 26 | Inner Lip | 2/8 |
| 27 | Inner Lip | 6/8 |
| 28 | Left Eye Pupil | 0/1 |
| 29 | Right Eye Pupil | 0/1 |

Table 10: Mapping of 300W's 68 landmarks to their face parts and Face Part-Anchored Landmark Positions (FPALPs).

| Landmark ID | Face Part | FPALP | Landmark ID | Face Part | FPALP |
|---|---|---|---|---|---|
| 1 | Face Contour | 0/32 | 35 | Nose Boundary | 4/6 |
| 2 | Face Contour | 2/32 | 36 | Nose Boundary | 5/6 |
| 3 | Face Contour | 4/32 | 37 | Left Eye | 0/6 |
| 4 | Face Contour | 6/32 | 38 | Left Eye | 1/6 |
| 5 | Face Contour | 8/32 | 39 | Left Eye | 2/6 |
| 6 | Face Contour | 10/32 | 40 | Left Eye | 3/6 |
| 7 | Face Contour | 12/32 | 41 | Left Eye | 4/6 |
| 8 | Face Contour | 14/32 | 42 | Left Eye | 5/6 |
| 9 | Face Contour | 16/32 | 43 | Right Eye | 0/6 |
| 10 | Face Contour | 18/32 | 44 | Right Eye | 1/6 |
| 11 | Face Contour | 20/32 | 45 | Right Eye | 2/6 |
| 12 | Face Contour | 22/32 | 46 | Right Eye | 3/6 |
| 13 | Face Contour | 24/32 | 47 | Right Eye | 4/6 |
| 14 | Face Contour | 26/32 | 48 | Right Eye | 5/6 |
| 15 | Face Contour | 28/32 | 49 | Outer Lip | 0/12 |
| 16 | Face Contour | 30/32 | 50 | Outer Lip | 1/12 |
| 17 | Face Contour | 32/32 | 51 | Outer Lip | 2/12 |
| 18 | Left Eyebrow | 0/9 | 52 | Outer Lip | 3/12 |
| 19 | Left Eyebrow | 1/9 | 53 | Outer Lip | 4/12 |
| 20 | Left Eyebrow | 2/9 | 54 | Outer Lip | 5/12 |
| 21 | Left Eyebrow | 3/9 | 55 | Outer Lip | 6/12 |
| 22 | Left Eyebrow | 4/9 | 56 | Outer Lip | 7/12 |
| 23 | Right Eyebrow | 0/9 | 57 | Outer Lip | 8/12 |
| 24 | Right Eyebrow | 1/9 | 58 | Outer Lip | 9/12 |
| 25 | Right Eyebrow | 2/9 | 59 | Outer Lip | 10/12 |
| 26 | Right Eyebrow | 3/9 | 60 | Outer Lip | 11/12 |
| 27 | Right Eyebrow | 4/9 | 61 | Inner Lip | 0/8 |
| 28 | Nose Bridge | 0/3 | 62 | Inner Lip | 1/8 |
| 29 | Nose Bridge | 1/3 | 63 | Inner Lip | 2/8 |
| 30 | Nose Bridge | 2/3 | 64 | Inner Lip | 3/8 |
| 31 | Nose Bridge | 3/3 | 65 | Inner Lip | 4/8 |
| 32 | Nose Boundary | 1/6 | 66 | Inner Lip | 5/8 |
| 33 | Nose Boundary | 2/6 | 67 | Inner Lip | 6/8 |
| 34 | Nose Boundary | 3/6 | 68 | Inner Lip | 7/8 |

Table 11: Mapping of WFLW's 98 landmarks to their face parts and Face Part-Anchored Landmark Positions (FPALPs).

| Landmark ID | Face Part | FPALP | Landmark ID | Face Part | FPALP |
|---|---|---|---|---|---|
| 1 | Face Contour | 0/32 | 50 | Right Eyebrow | 7/9 |
| 2 | Face Contour | 1/32 | 51 | Right Eyebrow | 8/9 |
| 3 | Face Contour | 2/32 | 52 | Nose Bridge | 0/3 |
| 4 | Face Contour | 3/32 | 53 | Nose Bridge | 1/3 |
| 5 | Face Contour | 4/32 | 54 | Nose Bridge | 2/3 |
| 6 | Face Contour | 5/32 | 55 | Nose Bridge | 3/3 |
| 7 | Face Contour | 6/32 | 56 | Nose Boundary | 1/6 |
| 8 | Face Contour | 7/32 | 57 | Nose Boundary | 2/6 |
| 9 | Face Contour | 8/32 | 58 | Nose Boundary | 3/6 |
| 10 | Face Contour | 9/32 | 59 | Nose Boundary | 4/6 |
| 11 | Face Contour | 10/32 | 60 | Nose Boundary | 5/6 |
| 12 | Face Contour | 11/32 | 61 | Left Eye | 0/8 |
| 13 | Face Contour | 12/32 | 62 | Left Eye | 1/8 |
| 14 | Face Contour | 13/32 | 63 | Left Eye | 2/8 |
| 15 | Face Contour | 14/32 | 64 | Left Eye | 3/8 |
| 16 | Face Contour | 15/32 | 65 | Left Eye | 4/8 |
| 17 | Face Contour | 16/32 | 66 | Left Eye | 5/8 |
| 18 | Face Contour | 17/32 | 67 | Left Eye | 6/8 |
| 19 | Face Contour | 18/32 | 68 | Left Eye | 7/8 |
| 20 | Face Contour | 19/32 | 69 | Right Eye | 0/8 |
| 21 | Face Contour | 20/32 | 70 | Right Eye | 1/8 |
| 22 | Face Contour | 21/32 | 71 | Right Eye | 2/8 |
| 23 | Face Contour | 22/32 | 72 | Right Eye | 3/8 |
| 24 | Face Contour | 23/32 | 73 | Right Eye | 4/8 |
| 25 | Face Contour | 24/32 | 74 | Right Eye | 5/8 |
| 26 | Face Contour | 25/32 | 75 | Right Eye | 6/8 |
| 27 | Face Contour | 26/32 | 76 | Right Eye | 7/8 |
| 28 | Face Contour | 27/32 | 77 | Outer Lip | 0/12 |
| 29 | Face Contour | 28/32 | 78 | Outer Lip | 1/12 |
| 30 | Face Contour | 29/32 | 79 | Outer Lip | 2/12 |
| 31 | Face Contour | 30/32 | 80 | Outer Lip | 3/12 |
| 32 | Face Contour | 31/32 | 81 | Outer Lip | 4/12 |
| 33 | Face Contour | 32/32 | 82 | Outer Lip | 5/12 |
| 34 | Left Eyebrow | 0/9 | 83 | Outer Lip | 6/12 |
| 35 | Left Eyebrow | 1/9 | 84 | Outer Lip | 7/12 |
| 36 | Left Eyebrow | 2/9 | 85 | Outer Lip | 8/12 |
| 37 | Left Eyebrow | 3/9 | 86 | Outer Lip | 9/12 |
| 38 | Left Eyebrow | 4/9 | 87 | Outer Lip | 10/12 |
| 39 | Left Eyebrow | 5/9 | 88 | Outer Lip | 11/12 |
| 40 | Left Eyebrow | 6/9 | 89 | Inner Lip | 0/8 |
| 41 | Left Eyebrow | 7/9 | 90 | Inner Lip | 1/8 |
| 42 | Left Eyebrow | 8/9 | 91 | Inner Lip | 2/8 |
| 43 | Right Eyebrow | 0/9 | 92 | Inner Lip | 3/8 |
| 44 | Right Eyebrow | 1/9 | 93 | Inner Lip | 4/8 |
| 45 | Right Eyebrow | 2/9 | 94 | Inner Lip | 5/8 |
| 46 | Right Eyebrow | 3/9 | 95 | Inner Lip | 6/8 |
| 47 | Right Eyebrow | 4/9 | 96 | Inner Lip | 7/8 |
| 48 | Right Eyebrow | 5/9 | 97 | Left Eye Pupil | 0/1 |
| 49 | Right Eyebrow | 6/9 | 98 | Right Eye Pupil | 0/1 |

