# OpenReview forum: "Towards Unified Dynamic Face Landmark Detection"
_ICLR.cc/2026/Conference — Submitted to ICLR 2026_

### Official Review · Reviewer_J6iZ · 2025-10-29

**Soundness:** 3
**Presentation:** 3
**Contribution:** 3
**Rating:** 8
**Confidence:** 5

**Summary:**

This paper addresses two fundamental limitations in face landmark detection (FLD): the need to train separate models for each "N-point" dataset and the inability of these models to output a flexible number of landmarks. It introduces a paradigm-shifting solution built on two core innovations. First, it proposes Face Part-Anchored Landmark Positions (FPALPs), a novel, universal representation that defines each landmark not by its coordinates, but by its normalized, semantic position along a face-part contour. Second, it presents a query-based detection framework that accepts FPALP-based queries to enable fully dynamic, on-demand landmark prediction.

This new paradigm successfully enables a single model to be trained on a unified fusion of heterogeneous datasets, and to dynamically predict any number of landmarks at inference time. Experiments demonstrate that this approach achieves these critical new capabilities while maintaining performance that is highly competitive with specialized, state-of-the-art methods.

Crucially, the paper implicitly surfaces a fundamental challenge in the FLD task itself: the subtle inconsistencies in how different datasets define the precise contour shapes of face parts. The model's need to learn an "average" representation from these slightly varied annotations likely explains the minor performance gap compared to models trained on a single, self-consistent dataset. This is not a flaw of the method, but rather an inherent property of the data that this unified approach is the first to successfully navigate.

**Strengths:**

1. This paper introduces a powerful conceptual shift for the field of face landmark detection (FLD). By proposing the Face Part-Anchored Landmark Positions (FPALPs) representation, it reframes landmark detection from a dataset-specific coordinate regression problem into a generalized, semantic-querying task. This is a highly elegant and impactful innovation that addresses long-standing issues of data fragmentation and model inflexibility.
2. The FPALP concept directly unlocks three highly valuable capabilities that were previously unattainable with a single model:
a) Unified Multi-Dataset Training: It provides a principled framework for combining heterogeneous "N-point" datasets, significantly increasing the volume and diversity of data available for training a single, more robust model.
b) Flexible, On-Demand Prediction: The query-based architecture allows for dynamic prediction of any number of landmarks at inference time, offering unprecedented flexibility for diverse downstream applications.
3. The paper demonstrates, through rigorous experiments, that its unified and dynamic framework achieves performance that is highly competitive with specialized, state-of-the-art models. The fact that it incurs only a negligible performance drop while providing enormous gains in data utilization and output flexibility is a remarkable achievement and a testament to the soundness of the proposed approach.

**Weaknesses:**

The unified training approach, by design, exposes the model to subtle but real inconsistencies in how different datasets define the exact contour shape of a given face part. This introduces a form of unavoidable "label noise." The model's need to learn a generalized, "average" representation of each contour to accommodate this variance is likely the primary reason its performance, while highly competitive, does not exceed that of a specialist model trained on a single, perfectly self-consistent dataset. This is a fundamental trade-off between specialization and generalization, and the paper makes a compelling case for the immense value gained on the generalization front.

**Questions:**

1. Your results show that the unified model is highly competitive, yet does not strictly surpass specialist SOTA models on their native datasets. We believe this is not a weakness. Would you agree that this highlights a fundamental trade-off between generalization (from diverse data) and specialization (on self-consistent data)? Framing your results in this light seems to strengthen, rather than weaken, your contribution, as it showcases the immense flexibility gained for a minimal cost in specialization.
2. Following up on that, you astutely identified an average intra-cluster distance of 2.22 pixels when unifying the templates. Could one interpret this value as a proxy for the inherent "label noise" or "ambiguity" that exists between datasets? If so, would it be fair to say that your model, by learning a robust average representation, is performing optimally under this noisy supervision, and this inherent ambiguity itself constitutes the performance ceiling compared to a specialist model?
3. Given that the core challenge you've surfaced is this inherent ambiguity in contour definitions, have you considered future work that models this uncertainty explicitly? For instance, instead of predicting a single coordinate (x, y) for a given FPALP query, could the framework be extended to predict a probability distribution or an uncertainty ellipse? This seems like a natural and exciting next step to create models that are not only unified but also aware of the data's intrinsic ambiguities.

---

> ### Author Response · Authors · 2025-11-17
> **Rebuttal to Reviewer J6iZ**
>
> We thank the reviewer for the highly insightful and positive assessment of our work. We appreciate the recognition that FPALPs reformulate FLD into a semantic, query-driven task and that unified training across heterogeneous templates naturally introduces the trade-off between specialization and generalization.
>
> \
> **1. Performance trade-off between generalization and specialization**
>
> We agree and resonate with your interpretation: specialist models benefit from perfectly self-consistent annotation regimes, whereas a unified model must accommodate the subtle but genuine annotation variations across datasets. The minor performance gap on native test sets is therefore expected and reflects an inherent property of the data, not a limitation of FPALPs or the unified architecture. In contrast, the unified model gains the ability to train on substantially more diverse samples and to generalize across templates and landmark definitions—capabilities that specialist formulations cannot provide.
>
> **2. Interpretation of the intra-cluster distance**
>
> We agree with you and further theorize that this value quantifies a blend of (1) semantic positioning inconsistency across multiple poses, arising from differences in how datasets define contour trajectories under foreshortening, profile rotations, occlusions and different facial expressions, and (2) subjective annotation noise at the annotator level. FPALPs operate under this noisy supervision yet still learn a stable “average” contour representation, and we believe that our unified results are near the ceiling imposed by these inherent ambiguities. A promising direction for future work is to explicitly model and decompose this noise, e.g., by estimating semantic curve variability separately from annotator-level deviations and training the model to minimize the former while remaining robust to the latter.
>
> This also motivates an additional research direction: introducing an optional dataset-type token to allow the model to encode dataset-specific annotation biases. This could enable the model to internally separate stylistic annotation variations (e.g., differing definitions of facial boundaries across datasets) from the underlying geometric structure, and even facilitate annotation denoising by comparing dataset-conditioned predictions against a unified, dataset-agnostic representation.
>
> **3. Explicitly modeling the uncertainty in landmark positions as future work**
>
> We appreciate your suggestion to predict distributions instead of single point estimates. Extending FPALP queries to output uncertainty ellipses, heatmap-based spatial densities, or probabilistic contour segments is indeed a compelling next step, and we are grateful for your insight in highlighting this direction. Your observation aligns closely with a natural evolution of our representation: the 2D FPALP formulation discussed in Appendix A.8 can be viewed as a higher-dimensional extension of the same idea, where the model no longer parameterizes only a 1D curve but instead defines a continuous 2D semantic region. Under such a formulation, the same probabilistic output mechanisms you suggest could be expressed as spatial uncertainty surfaces over these regions, allowing the model to capture fine-grained ambiguity arising from annotation variance or pose distortions.
>
> ---
> We sincerely appreciate your thoughtful analysis and the depth of understanding reflected in your review. Your framing of the generalization–specialization trade-off, your interpretation of the inherent annotation ambiguity, and your forward-looking suggestions meaningfully complement the goals of our work. We are glad that our formulation resonated with your insights, and we thank you again for your supportive assessment and for highlighting avenues that can further strengthen and extend FPALP-based landmark reasoning in future research. Please let us know if there is anything else we can do to help you further improve your evaluation of our work.

---

> ### Author Response · Authors · 2025-11-25
>
> Dear Reviewer J6iZ,
>
> Please let us know if our rebuttal adequately addressed your concerns or if there is anything else we can clarify. We appreciate your time and consideration.
>
> *Thank You,*
>
> *All Authors*

---

### Official Review · Reviewer_NXcn · 2025-10-30

**Soundness:** 3
**Presentation:** 3
**Contribution:** 2
**Rating:** 4
**Confidence:** 3

**Summary:**

This paper proposes a unified landmark representation, FPALP, and a corresponding dynamic landmark detection network built upon it. FPALP is defined based on the ratio of lengths along pre-defined facial part curves. The proposed detection framework consists of three components: (1) Image-Agnostic Landmark Encoding Generation, which produces embeddings for both facial parts and their ratio representations; (2) Landmark Query Initialization, which interacts with image features through an attention mechanism; and (3) Landmark Query Refinement, which decodes the extracted features into the final landmark coordinates. The framework can be trained on datasets with varying numbers of annotated landmarks rather than relying on datasets with a fixed number. Extensive experiments demonstrate that the method achieves competitive accuracy on individual datasets and superior performance in cross-dataset evaluations.

**Strengths:**

1.	The paper introduces FPALP, a unified landmark representation defined by evenly distributed points along facial part curves. This design enables the framework to be trained across multiple datasets (e.g., WFLW, 300W, AFLW-19) while producing outputs that are not constrained to a fixed number of landmarks.
2.	The paper proposes a dynamic landmark detection framework based on FPALP, which predicts landmark positions using a fusion strategy that combines facial part embeddings and image features through a Dot-Product Attention Map and Deformable Image Cross-Attention.
3.	Extensive experiments demonstrate strong cross-dataset performance, and the ablation study is thorough, carefully examining both the image encoder and text encoder components.

**Weaknesses:**

Major:
1. The paper lacks experiments evaluating the accuracy of facial part curve prediction compared to interpolation-based methods. Since the core idea of FPALP is to model and predict facial part curves, such comparisons are essential to demonstrate FPALP’s effectiveness. The absence of these results leaves the validity of FPALP’s key contribution somewhat uncertain.
2. The paper does not adequately address the misalignment issue across datasets. One of the underlying assumptions is that the landmarks in each dataset are evenly distributed along facial part curves; however, this is not strictly true for all datasets. Moreover, FPALP’s handling of the start and end points of each curve— which vary across datasets—remains unclear. Although the authors briefly discuss this limitation, the analysis lacks sufficient depth.
3. The qualitative results focus primarily on frontal faces, with limited examples involving large poses. This raises concerns about the stability and robustness of FPALP under challenging geometric variations.

Minor:
1. Some parts of the mathematical formulation in the method section are unclear. For instance, the meaning of H_T in R^((H_T×W_ei×L)) (line 253) is not explained—possibly a typo? The paper would benefit from more explicit symbol definitions or improved figure annotations for clarity.
2. The use of text descriptions for facial part phrases appears unnecessary given the small number of facial parts. Although the authors attempt to justify this choice in the ablation study by replacing the frozen text encoder with a learnable embedding, the mechanism of the learnable embedding is not sufficiently explained. If it is simply a non-frozen version of the text encoder, the benefit of textual representation becomes questionable. Using one-hot encodings for facial parts might be a simpler and equally effective alternative.

**Questions:**

See the weakness

---

> ### Author Response · Authors · 2025-11-17
> **Rebuttal to Reviewer NXcn - Part 1**
>
> We sincerely thank the reviewer for their thorough review and insightful feedback on our submission. We are very pleased that you recognized the core strengths of our work, particularly the introduction of FPALP as a unified landmark representation, our dynamic landmark detection framework, and the extensive experimental validation demonstrating strong cross-dataset performance. Below, we provide clarifications and responses to the main points of concern.
>
> \
> **1. Evaluating FPALP accuracy vs. interpolation-based curves**
>
> FPALPs are not designed to replicate interpolation-based curve generation, but to provide a semantic, dataset-aligned representation that unifies heterogeneous N-point templates without requiring dense ground-truth curves. Interpolation assumes uniformly sampled boundaries, whereas datasets like 300W, WFLW, and AFLW-19 contain semantically defined (and often unevenly spaced) anchor points. As discussed in Appendix A.2, *interpolation density varies naturally with facial geometry (e.g., tapered chin regions or curvature along the jawline) and with pose changes (e.g., density “compresses” in profile views and expands in frontal views due to foreshortening).* This makes direct interpolation-based comparisons ambiguous rather than measuring the representational validity of FPALPs. Evaluating FPALPs against interpolated curves would therefore measure adherence to uniform coordinate spacing—an artifact of the interpolation process—rather than the semantic spacing that reflects the true quality and purpose of a unified representation.
>
> Instead, we evaluate FPALP effectiveness using tests that directly probe whether the model has learned a stable geometric representation of each facial part. First, our dynamic landmark predictions at *multiple granularities* (Fig. 6, Sec. 4.3) consistently produce *ordered, smooth, and curvature-preserving trajectories* along facial part boundaries, showing that FPALPs provide a continuous and well-behaved parameterization of these curves. Second, when the training and evaluation templates differ, as examined in the *cross-template* ablations in Tables 3, 4 and 6b, the model must operate across datasets with differing or partially disjoint landmark definitions, yet it maintains competitive accuracy *without any interpolation-based alignment.* Together, these results demonstrate that FPALPs support consistent reasoning about face-part geometry across heterogeneous templates and datasets, without relying on explicit dense-curve supervision.
>
> **2. Handling Dataset Misalignment and Start/End Points**
>
> We acknowledge that our method operates under an approximate alignment of face templates when unifying diverse datasets. The three standard benchmarks (AFLW, WFLW, and 300W), used throughout prior FLD literature, indeed adopt different face templates, but their landmarks are consistently defined as evenly distributed semantic points along face-part curves (Fig. 2a). Our unified training demonstrates strong resilience to this approximate alignment: empirical results show only minor drops relative to specialist models while providing significantly *greater data utilization and output flexibility*, as also highlighted by Reviewer J6iZ.
>
> The FPALP formulation normalizes progression along each face part from 0 to 1 in a *dataset-agnostic manner.* Crucially, the start and end points of a face-part curve in FPALPs *are not fixed by any dataset*; they are defined once by the practitioner when specifying the face-part phrase and its member landmarks. FPALPs then assign each (face part, landmark) pair a normalized position in [0,1] along that user-defined ordering. This means that the *same physical landmark can legitimately receive different FPALP values under different, possibly overlapping, face-part phrases (e.g., “nose” = “nose bridge + left + right boundary of nose” vs “right nasal region” = “right boundary of nose + right nasolabial fold”)*, and differences in how individual datasets choose their “first” or “last” landmark on a contour *do not constrain the unified representation.* Appendix A.6 (“Detailed FPALP Formulation”) (and the implementation details in the appendix) provide step-by-step details on how landmarks from each considered dataset are mapped to FPALPs, effectively abstracting away dataset-specific ordering or endpoint conventions.
>
> Furthermore, as noted in Appendix A.3 (lines 702–705), we intentionally exclude datasets with severe annotation inconsistencies, since they negatively impact learning stability. This pragmatic design ensures that FPALPs remain both flexible—supporting multiple, overlapping, task-specific face parts—and robust to reasonable template misalignments, while maintaining high-quality representations across all parts.

---

> > ### Author Response · Authors · 2025-11-17
> > **Rebuttal to Reviewer NXcn - Part 2**
> >
> > **3. Addition of Large Pose Samples in Qualitative Results**
> >
> > We understand your concern about the representation of large pose examples in our qualitative results. To address this, we added Figures 8 and 9 in Appendix A.13, which showcase our system's dynamic landmark prediction capabilities on challenging occlusion and extreme pose cases from the WFLW test set. These examples demonstrate our model's stability and robustness in situations involving significant geometric variations, illustrating its ability to accurately reason and predict landmark positions even under such difficult conditions.
> >
> > **4. Typo Fix in Mathematical Formulation**
> >
> > Thank you for pointing out the notation type of $H_T$. It should have been $H_{E_I}$, consistent with $(H_{E_I}, W_{E_I})$ representing the spatial resolution of the image features. We have corrected this in the revised version.
> >
> > **5. Necessity of Text Descriptions for Facial Part Phrases**
> >
> > The “learnable embeddings” refer to standard trainable matrices $W_E \in R ^ {P \times d}$, where $P$ is the number of considered face parts and $d$ is the embedding dimension.
> >
> > Text embeddings from a pretrained text encoder capture rich semantic information and relational context about facial parts (e.g., the spatial and interactive relationship between "upper lip" and "lower lip" during speech or laughter or how an "eyebrow" contributes to an expression). While a learnable embedding might capture some feature distinctiveness, *it does not leverage the pre-existing knowledge about facial anatomy and inter-part relationships* that a robust text encoder provides, which is crucial for our framework's interpretability and generalization capabilities.
> >
> > Notably, only a text encoder allows users to *introduce new face-part descriptions or define custom semantic regions* (e.g., “inner brow ridge,” “left chin corner,” or “upper nasolabial fold”), since these phrases naturally embed into the same semantic space as the original parts. One-hot vectors or randomly initialized learnable embeddings cannot generalize to unseen face-part descriptions, and thus cannot inherit the inductive biases captured in pretrained text models.
> >
> > Our experimental findings (L384–407) consistently demonstrate that pretrained text encoders offer superior performance, faster convergence than learnable embeddings, and future-proof flexibility for describing additional user-specified face-part phrases that immediately benefit from the facial semantics encoded in the existing part representations.
> >
> > ---
> > We are grateful for your thoughtful review and for highlighting the strengths of our unified FPALP formulation, the soundness of the dynamic prediction framework, and the strong cross-dataset results. We hope the clarifications above address your remaining concerns regarding curve evaluation, template alignment, and the role of text embeddings. Please let us know whether we have addressed your concerns and whether you have any other questions or requests that would help you raise your score in favor of acceptance.

---

> > > ### Comment · Reviewer_NXcn · 2025-11-25
> > > **Improve my rating**
> > >
> > > I have read the author's rebuttal, and they have addressed most of my concerns. Although I still have some doubts regarding the practical applicability of the proposed FPALP-based landmark representation, the overall idea remains novel. Therefore, I have decided to raise my score to 6.

---

> > > > ### Author Response · Authors · 2025-11-25
> > > > **Rating Improvement Acknowledgement and Reviewer NXcn Rebuttal Summary**
> > > >
> > > > Thank you for taking the time to re-evaluate our submission and for raising your score to 6. We appreciate your acknowledgment that our rebuttal addressed most of your concerns. As outlined in our responses, we clarified the role of FPALPs relative to interpolation-based methods, detailed how start/end-point inconsistencies are handled through the normalized progression formulation, and expanded on large-pose qualitative results, text-encoder justification, and dataset-template alignment. We understand your remaining reservations about the practical applicability of FPALPs and would be glad to further clarify any aspect of the formulation, implementation, or evaluation. Please feel free to let us know if you have additional questions—we are happy to elaborate wherever needed.
> > > >
> > > > *Thank You,*
> > > >
> > > > *All Authors*

---

### Official Review · Reviewer_yDUK · 2025-10-31

**Soundness:** 2
**Presentation:** 2
**Contribution:** 2
**Rating:** 4
**Confidence:** 3

**Summary:**

This paper proposes a unified and dynamic face landmark detection (FLD) framework based on Face Part-Anchored Landmark Positions (FPALPs), which represent landmarks as normalized progression values along semantic face part contours. The method enables training on multiple "N-point" datasets (e.g., AFLW, 300W, WFLW) simultaneously and supports dynamic, on-demand landmark prediction at inference. A cross-modality decoder refines landmark queries constructed from FPALPs and text embeddings. Experiments show competitive performance with state-of-the-art methods while offering enhanced flexibility and generalization.

**Strengths:**

1. Unified Representation: The FPALP formulation effectively aligns heterogeneous landmark annotations across datasets, enabling a single model to learn from multiple sources without manual interpolation or 3D priors.
2. Dynamic Inference: The framework supports arbitrary landmark queries at test time, allowing customizable output granularity and adaptability to diverse downstream tasks—a significant advantage over fixed-output models.

**Weaknesses:**

1. Template Alignment Sensitivity: The method relies on approximate alignment of face templates across datasets, which may limit scalability when integrating new datasets with highly divergent landmark definitions or severe annotation inconsistencies.
2. Performance Trade-offs: While competitive overall, the model exhibits a slight performance drop on certain subsets (e.g., WFLW68) when trained on fused datasets, suggesting sensitivity to dataset-specific challenges and sampling strategies.

**Questions:**

1. How does the framework handle face parts that are heavily occluded or not present in the training data? The paper mentions future extensions but does not evaluate robustness under such conditions.
2. Could the dependency on text encoders (e.g., SentenceBERT) introduce biases or limit performance in low-resource languages or domains where facial part semantics differ?

---

> ### Author Response · Authors · 2025-11-17
> **Rebuttal to Reviewer yDUK - Part 1**
>
> We would like to thank the reviewer for their thorough review and valuable feedback on our submission. We are pleased that you recognized the core strengths of our work, particularly the unified representation through FPALPs, which effectively aligns heterogeneous landmark annotations without interpolation or 3D priors, and the dynamic inference capability, offering significant advantages over fixed-output models. We address your concerns and questions below.
>
> \
> **1. Template Alignment Sensitivity**
>
> We acknowledge that our method relies on an approximate alignment of face templates across datasets. As discussed in Section 5 ("Limitations") and Appendix A.11, the current framework demonstrates resilience to this approximate alignment. Our model, trained on diverse templates (19, 68, and 98 points), has shown that *only an approximate alignment is necessary for effective face part curve learning and is tolerant to reasonable misalignments.* We note that datasets with severely diverging landmark definitions and annotation inconsistencies do not contribute towards effective landmark generalization and pose a threat to landmark consistency, prediction stability, and robustness that may be critical for general downstream applications. As an example, in Appendix A.3 (lines 702-705), we explicitly noted our decision to exclude the COFW dataset from the unified training set due to its poor annotation quality, which led to overall performance degradation and adversely impacted the prediction of dense landmarks on all other datasets.
>
> **2. Performance Trade-offs**
>
> You correctly pointed out the slight performance drop on certain subsets, specifically WFLW68. As detailed in Appendix A.10, this is indeed a consequence of our unified training strategy. By optimizing for overall generalization across a wider variety of "N-point" datasets, the model's exposure to extremely challenging, dataset-specific cases (like those in WFLW68) can be comparatively reduced. However, as Reviewer J6iZ also noted, this trade-off enables our model *to achieve better landmark generalization* across multiple datasets and to handle a broader range of N-point formats, which is a *key contribution* of our work.

---

> ### Author Response · Authors · 2025-11-17
> **Rebuttal to Reviewer yDUK - Part 2**
>
> **3. Handling undefined and occluded face parts**
>
> In Appendix A.7 ("Handling Undefined or Occluded Face Parts") of our original manuscript, we elaborate on how our framework could be extended to address these challenging scenarios. We expand the section here for convenience:
>
>   1. We can utilize the text encoder to parse face part descriptions into latent embeddings that can be aligned with image features. Soft spatial attention maps based on the introduced face parts can be used to approximate the boundaries of unseen face parts, even under occlusion. Such an extension would enable the model to infer FPALP-like progression values for novel regions by projecting the learned attention map onto surrounding anchor contours. Additionally, a dynamic part discovery module could be trained using contrastive losses to bind new textual descriptions to consistent visual patterns across samples. This could potentially enable open-vocabulary part generalization in FLD, which could be an exciting avenue for future work.
>
>   2. We can also leverage visibility annotations per landmark, such as those provided in the MERL-RAV [1] dataset, to supervise the model in learning to selectively ignore occluded regions during training. This allows the framework to learn robust part representations even when portions of the face are not visible. Additionally, these visibility flags can be used to guide a gating mechanism or soft-attention masking module that modulates the contribution of occluded regions in the query or image features during inference, improving landmark prediction reliability under occlusion.
>
> We also included additional visualizations of challenging occlusion cases in Figure 8 of Appendix A.13, demonstrating our model's current ability to reason and predict landmarks despite partial and complete occlusions.
>
> **4. Limitations of including the text encoder in low-resource language or differing facial semantics settings**
>
> Thank you for raising the concern regarding the generalizability of our approach across languages and cultural contexts. We acknowledge that the choice of the training dataset and the text encoder could introduce biases or limitations when face part phrases are described using low-resource languages. In order to mitigate such biases and limitations, the definition of face part phrases should be standardized for a consistent interpretation across the different languages, and the text encoder would either need to be trained or fine-tuned on the target languages. While this is a crucial consideration for real-world deployment in a global context, our current work limits its scope to the English language context, deferring inter-linguistic adaptations and broader cross-cultural considerations to future research.
>
> ---
>
> We appreciate your thoughtful analysis and the opportunity to clarify these aspects of our work. We hope our detailed responses regarding template alignment, unified-training trade-offs, and extensions for occlusion and multilingual settings address your concerns and highlight the strengths you already identified—namely, the unified FPALP representation and dynamic inference capabilities. Please let us know whether we have addressed your concerns and whether you have any other questions or requests that would help you raise your score in favour of acceptance.
>
> ---
>
> [1] Kumar, Marks, and Mou et al., Luvli face alignment: Estimating landmarks’ location, uncertainty, and visibility likelihood. In CVPR, 2020.

---

> ### Author Response · Authors · 2025-11-25
>
> Dear Reviewer yDUK,
>
> Please let us know if our rebuttal adequately addressed your concerns or if there is anything else we can clarify. We appreciate your time and consideration.
>
> *Thank You,*
>
> *All Authors*

---

### Official Review · Reviewer_8hur · 2025-11-01

**Soundness:** 3
**Presentation:** 3
**Contribution:** 3
**Rating:** 6
**Confidence:** 3

**Summary:**

This paper proposed Face Part-Anchored Landmark Positions as a generalized representation for landmark, w.r.t. face part's contour.
This enables test-time specification of landmarks, with a single model (backbone and head) trained on any N-point datasets.
Experiments prove this method generalizes while keep competitive result with existing SOTAs.

**Strengths:**

Generalization to arbitrary landmark definitions is an important problem for the community. FPALP is a sound solution, and the authors also conducted extensive experiments to validate its effectiveness. Also, the paper is well-written and easy to follow.

**Weaknesses:**

As you mentioned in Sec 4 and A.10, WFLW dataset consists of extreme poses and occlusions. In A.7 you said those two factors can be challenging for your FPALP. It will be good if you can show some of those cases in addition to Figure 7.

**Questions:**

1. Fig. 3 the meaning of "Face Contour" is unclear.
2. Is FPALP limited to contour landmarks in theory? For example, can we extend this method to represent a landmark at the center of eyeball?
3. As you mentioned at Line 242, if you believe facial layout semantics within text encoder is important, it will be helpful to show results where FPALP is trained with text encoder that was already aligned with image encoder, e.g., CLIP.

---

> ### Author Response · Authors · 2025-11-17
> **Rebuttal to Reviewer 8hur**
>
> We would like to thank the reviewer for their time in reviewing our work and providing thoughtful feedback. We are glad that you found our work to be valuable, recognizing the importance of generalization to arbitrary landmark definitions, the soundness of our FPALP solution, and the extensive experimental validation. We also appreciate your positive comments on the paper's clarity and readability. We address your concerns and questions below.
>
> \
> **1. Addition of Large Pose Samples in Qualitative Results**
>
> Thank you for asking us to add additional visualizations pertaining to extreme poses and occlusions. We understand the importance of showcasing our model's robustness in these challenging scenarios. We have added these visualizations via *Figure 8 and Figure 9 in Section A.13* of our appendix, demonstrating the system's performance on such cases from the WFLW test set.
>
> **2. Meaning of "Face Contour”**
>
> In Figure 3, "Face Contour" refers to one of the "Face Part Phrases" that are fed into the Text Encoder.  "Face Contour" is a fundamental face part curve, outlining the overall perimeter and shape of the face. It is treated as a distinct semantic entity alongside other parts like "left/right eye(brow)," "inner/outer lip," etc., allowing our model to understand and predict landmarks along this specific facial boundary. In Figure 3, we demonstrate our framework’s pipeline by requesting the model to predict the coordinates of the Face Contour’s start (FPALP 0.0), the middle (FPALP 0.5), and the end (FPALP 1.0) respectively.
>
> **3. Is FPALP limited to contour landmarks in theory?**
>
> The current FPALP formulation primarily uses a 1D progression value along face part curves or contours. While many landmarks are naturally located on such boundaries (e.g., along the jawline, lips, or eyebrows), the framework is not strictly limited to them, and indeed, we already apply it to non-contour landmarks. For instance, the center of the eyeball, which is not part of a contour, is mapped as a specific FPALP within the "left eye" or "right eye" face part (see Appendix Tables 8, 9, 10, and 11 for "Left Eye Pupil" and "Right Eye Pupil" landmarks). This is achieved by treating these isolated points as a face part with a single "progression value" (e.g., 0.0, as shown for pupils).
>
> **4. Use of text encoders with better image alignment**
>
> You raise a very pertinent and insightful point, and we appreciate you prompting us to elaborate further. We have indeed explored the use of text encoders explicitly designed for better alignment with facial images. Specifically, we experimented with the FaRL [1] text encoder (which we mention in Appendix A.4), built with the same architecture as CLIP ViT-B16, but crucially trained with facial images and their corresponding text captions from the LAION-FACE dataset. This training was intended to provide a more domain-specific alignment with facial imagery.
>
> However, while FaRL exhibited a strong general understanding of facial attributes due to its LAION-FACE training, we observed that its textual descriptions predominantly focused on general attributes like "smiling girl with party wig" or "the beautiful bride with the sunlight shining on her" (as noted in lines 394-397 of our main paper). These descriptions lacked the specific face-part intricacies required for our FPALP formulation, such as the nuanced semantics involved in "the squinting of the eyes and broadening of the lips during a laugh," or "interactions with makeup and accessories."
>
> In contrast, SentenceBERT, having been pretrained on diverse and extensive textual corpora, demonstrated a superior ability to effectively encode these more detailed and nuanced characteristics of individual face parts. While our study confirmed that using the FaRL text encoder yielded superior performance compared to generic learnable embeddings, our experiments ultimately revealed that SentenceBERT outperformed FaRL’s text encoder for our specific task. This indicated that, for *generating image-agnostic landmark encodings using face part phrases*, a superior representation of the specific semantics of face parts is achieved by *using a strong pretrained text encoder.* Therefore, despite the potential benefits of image-text alignment, the pretrained, lightweight SentenceBERT proved to be the more effective choice for encoding face part phrases in our framework.
>
> ---
>
> We sincerely appreciate your thoughtful questions and the constructive nature of your feedback. We hope our clarifications regarding the extended visualizations, the flexibility of FPALPs beyond contour-based landmarks, and our experiments using more image-aligned text encoders adequately address your concerns. Please let us know whether we have addressed your concerns and whether you have any other questions or requests that would help you raise your score.
>
> ---
> [1] Zheng and Yang et al., General facial representation learning in a visual-linguistic manner. In CVPR, June 2022.

---

> ### Author Response · Authors · 2025-11-25
>
> Dear Reviewer 8hur,
>
> Please let us know if our rebuttal adequately addressed your concerns or if there is anything else we can clarify. We appreciate your time and consideration.
>
> *Thank You,*
>
> *All Authors*

---

### Author Response · Authors · 2025-11-17
**General Response to All Reviewers (8hur, yDUK, NXcn, J6iZ)**

We thank all reviewers, 8hur, yDUK, NXcn, and J6iZ, for their careful evaluation and constructive feedback. Across the reviews, several shared strengths were noted. Reviewers highlighted the value of FPALPs as a unified semantic representation for heterogeneous N-point datasets, the ability to specify arbitrary landmarks at test time, and the empirical competitiveness of the proposed framework. Reviewer 8hur specifically emphasized that generalization to arbitrary landmark definitions is an important problem for the community, that FPALPs offer a sound and effective solution validated through extensive experiments. Reviewer J6iZ provided a detailed analysis of the conceptual impact of FPALPs, the positive implications of unifying multiple annotation templates, and how the approach exposes fundamental data-level ambiguities that shape the generalization–specialization trade-off.

The concerns raised were well-defined and distinct across reviewers.

- Reviewer 8hur asked about additional qualitative examples, clarification of the “face contour” phrase, applicability to non-contour points, and the role of image-aligned text encoders.
- Reviewer yDUK raised questions regarding template alignment sensitivity, performance trade-offs on WFLW-68, occlusion handling, and multilingual considerations.
- Reviewer NXcn requested deeper analysis around interpolation-based comparisons, start/end-point conventions across datasets, qualitative results under large pose, and justification for using text embeddings.
- Reviewer J6iZ focused on interpreting dataset inconsistencies, analyzing the intra-cluster distance in the context of inter-dataset label noise/ambiguity, and future extensions involving explicit uncertainty modeling.

We address each of these points individually in the following rebuttal sections. With these clarifications, and given the consistently identified strengths of the formulation, generalization ability, and empirical performance of our method, we respectfully hope that the reviewers will consider recommending acceptance of the work once their specific concerns have been addressed.

---

### Author Response · Authors · 2025-11-17
**Changes to the Paper**

Revision 1:

1) Following Reviewer NXcn's review, we fixed the notation typo in L253.
2) Following Reviewer 8hur and NXcn's recommendations, we added Figures 8 and 9 to the Appendix Section A.13 to demonstrate our model's ability to reason and predict landmarks under the challenging cases of occlusion and extreme pose cases respectively.
3) Following our reply to Reviewer 8hur, we added L728-734 in the Appendix Section A.4 to further detail and reason our usage of SentenceBERT vs the CLIP-based FaRL text encoder.
4) Following our reply to Reviewer yDUK, we added L699-705 in the Appendix Section A.2 to reflect the limitations of using the text encoder in low-resource language settings and mitigation strategies to improve the same.
5) Following our reply to Reviewer NXcn, we added L775-783 in the Appendix Section A.6 to clarify that our FPALP formulation is agnostic to landmark sequences from the source dataset and provides flexibility by design as landmark membership to face parts can be non-exclusive in nature.
6) Following our reply to Reviewer J6iZ, we added L864-871 in the Appendix Section A.11 to explain our analysis of the intra-cluster distance during the phase of the face template alignment of the source datasets.

---

### Meta-Review · Area_Chair_eZW5 · 2026-01-03

**Summary:**

This paper observes that different face landmark datasets use varied and incompatible ground-truth landmark definitions, making models trained on them difficult to generalize. To address this issue, the main contribution of the paper is a generalized landmark representation, termed Face Part–Anchored Landmark Positions (FPALPs). This representation uses contours associated with facial parts, where each contour is split into landmark points from different datasets using a global indexing scheme based on the ordering of landmarks from the start and end of the face-part contour (user-specified). The paper further proposes a query-based, image-agnostic landmark embedding to retrieve the desired landmarks for specific datasets, trained in a supervised setting on a larger dataset formed by combining individual datasets. Experiments on standard benchmarks show that the proposed approach achieves performance comparable to dataset-specific models.

**AC Comments:**

The paper received overall positive to mixed reviews, including two borderline accepts, one accept, and one borderline reject. Reviewers acknowledged the idea of unifying multiple incompatible face landmark annotations into the FPALPs format. However, concerns were raised on several fronts, namely: i) the lack of state-of-the-art results, with a slight drop in performance compared to specialist models; ii) limited insight into model behavior under occlusions; iii) lack of comparisons to interpolation-based methods; and iv) insufficient clarity on the progressive score mechanism.

In response, the authors acknowledged the trade-off between generalist and specialist models, which they argue leads to the observed performance drop relative to specialist approaches. They also presented new qualitative results demonstrating performance under occlusions and large pose variations, and provided clarification on how landmark indices are derived using the progressive scoring method along face-part contours. After the rebuttal, one reviewer raised their score to borderline accept.

The AC independently reviewed the paper and finds that most of the issues raised by the reviewers are addressed to some extent. However, there are additional concerns that may have been overlooked. Specifically:

1) The FPALPs format appears to function primarily as a global indexing scheme that combines landmark indices from multiple datasets, enabling training on a larger unified dataset composed of several face landmark datasets. In this context, it is unclear why performance would drop relative to specialist models. Is this degradation due to ambiguity in referring to dataset-specific landmarks introduced by the proposed indexing scheme? The paper appears to attribute this solely to generalist-versus-specialist trade-offs without providing a clear explanation. Moreover, given that the unified representation enables training on larger datasets, its lower-than-state-of-the-art performance raises important questions about the practical benefits of the proposed approach.

2) The role of the proposed query-based network architecture in determining performance is also unclear. It is not evident how a specialist model would perform if trained using the same architecture with FPALPs defined on a single dataset. For example, if one considers only the WFLW dataset and uses FPALPs indices specific to that dataset, how would performance compare to state-of-the-art approaches on WFLW? Such an experiment would help disentangle the contributions of training dataset size, the FPALPs representation, and architectural changes. The AC considers this an important missing ablation study.

3) As noted by the reviewers, model performance on occluded parts—despite those parts being specified in the queries—remains unclear. The paper provides only qualitative results, without quantitative analysis. This could be systematically evaluated using synthetic occlusions of landmarks or face parts, to assess whether the proposed unification scheme enables robust or dynamic prediction of occluded landmarks and how closely predictions match the occluded ground truth.

In summary, while the proposed contributions are incremental, the results—despite leveraging a large, unified dataset—do not achieve state-of-the-art performance, for reasons that remain insufficiently justified. Moreover, the paper lacks critical ablation studies analyzing the influence of training data size, architectural choices, and the proposed unified representation relative to comparable state-of-the-art approaches. These issues are important to address for acceptance.

**Reviewer Concerns:**

*Reviewer 8hur* points out that the presented approach may not work well under face landmark occlusions or extreme pose variations.

*Reviewer yDUK* argues that the unification methodology assumes approximate alignment of face landmarks across datasets, which could impact the generalizability of the proposed approach. The reviewer also notes that the method leads to a slight drop in performance on challenging benchmarks, suggesting sensitivity to dataset-specific issues. In addition, the reviewer questions how the method handles occlusions and whether reliance on pre-trained text encoders limits performance.

*Reviewer NXcn* points out the lack of comparisons to interpolation-based methods, the paper’s oversight of landmark imbalance along contours across datasets, the limited explanation of how progressive scores are assigned between the start and end of contours, and the focus of qualitative results on frontal poses while ignoring more challenging cases.

*Reviewer J6iZ* praises the approach, describing it as a powerful conceptual shift in face landmark detection. The reviewer raises a few minor concerns, including the trade-off between generalization and specialization that limits performance relative to specialist models, as well as the lack of consideration of uncertainty in model predictions.

**Reviewer Scores:**

**Reviewer 8hur:** The authors provide Figures 8 and 9 showing landmark detections under occlusions.

[*AC’s thoughts on the response*] The qualitative results presented during the rebuttal provide some level of confidence in addressing the reviewer’s concern, and the AC believes the reviewer may have maintained the score at “borderline accept.”

**Reviewer yDUK:** The authors agree with the reviewer’s criticisms that: i) their templates assume approximate alignment of landmarks across datasets, and ii) the approach may be disadvantaged by dataset-specific details when compared to generalization. Further, as expanded in Appendix A.7, the method currently cannot natively support occlusion handling, although it could potentially be extended using conditional parsing via text encoders; however, this is beyond the scope of the current unified setup. The authors also acknowledge that the use of pre-trained models may introduce biases.

[*AC’s take on the response*] The issues raised by the reviewer do not appear to be adequately addressed by the authors’ response. As such, the reviewer would likely have sustained the original score.

**Reviewer NXcn:** The authors argue that interpolation-based methods assume uniformly sampled boundaries, which may not hold for most datasets, and that their approach can regress contours correctly under varying granularities (Figure 6). They further explain how progressive scores are computed between the start and end points in Appendix A.6. The key idea appears to be unifying dataset-specific landmarks along face contours, assigning a unified, dataset-agnostic indexing scheme, and extracting landmarks via dataset-dependent queries. The authors also point to Figures 8 and 9 demonstrating robustness to large pose variations and occlusions. The reviewer is satisfied with the response and raises the score to “borderline accept.”

**Reviewer J6iZ:** The authors reiterate their generalization-versus-specialization argument discussed earlier and agree that incorporating uncertainty into the pipeline is a promising next step.

---

### Decision · Program_Chairs · 2026-01-26

Reject

---

> ### Public Comment · ~Varshanth_Rao1 · 2026-03-11
> **Response to the Decision and Meta-Review**
>
> We thank the Area Chair and reviewers for their time and feedback.
>
> While we appreciate the effort invested in the evaluation process, we believe several statements in the *meta-review do not accurately reflect the content of the paper and may lead to an incomplete and incorrect understanding of our contributions*. We therefore provide the following clarifications.
>
> **1) Performance relative to specialist models.**
> The meta-review states that the performance drop is attributed solely to a generic “generalist vs. specialist” explanation. This characterization does not reflect the discussion presented in the paper or the rebuttal. Our response to Reviewer J6iZ (who rated the paper 8 with confidence 5) provides *a detailed explanation of factors* that contribute to this behavior when unifying datasets with incompatible landmark definitions. Importantly, the goal of this work is not to outperform specialist models on individual benchmarks, but to enable unified and dynamic landmark prediction across datasets. We *explicitly emphasize* this in multiple locations in the manuscript *(TL;DR statement, L139–140, L340–342)*.
>
> We also note that *evaluating our method only through performance on individual dataset benchmarks assesses a different objective* than the one targeted by this work. Our contribution is the ability to perform unified and dynamic face landmark detection across datasets with incompatible annotations. *A more appropriate indicator of this capability is cross-dataset evaluation*, where our method outperforms existing SOTA approaches. Moreover, our framework *enables near zero-shot evaluation* on previously unseen landmark definitions, a capability that existing specialist models *cannot* support.
>
> **2) Role of the architecture and representation.**
> The meta-review suggests that the paper lacks an ablation isolating the role of FPALPs when trained on a single dataset. *This experiment is already included in the main manuscript.* Specifically, training and evaluating the model on WFLW using FPALPs defined for that dataset is reported in Table 4 (Row 2, Column 4), which can be directly compared with specialist methods reported in Table 2 (Column 6). These results provide the exact comparison requested in the meta-review.
>
> **3) Evaluation under occlusions.**
> During rebuttal we provided additional qualitative examples demonstrating behavior under occlusion and large pose variation, which addressed Reviewer NXcn’s concern and resulted in an increased score. We also report results on the 300W Challenging split in Table 2, following standard evaluation practice used in prior FLD work. While dedicated occlusion-specific benchmarks could provide additional insight, such evaluations are largely orthogonal to the primary objective of this work, which is to enable unified and dynamic landmark prediction across datasets with incompatible annotations. Constructing such a benchmark across multiple datasets would require substantial additional effort, including recreating experiments for prior methods whose implementations are not publicly available. Moreover, no prior FLD work evaluated under such a protocol. For consistency and comparability, we therefore follow the standard evaluation protocols used in the existing FLD literature.
>
> In summary, the contribution of this work is the introduction of a unified landmark representation that enables a single model to operate across datasets with incompatible annotations and dynamically query different landmark configurations. We are encouraged that *three of the four reviewers recommended acceptance* despite the method not being designed to optimize benchmark-specific performance. We appreciate the feedback and hope this work contributes to advancing unified face landmark detection.